# PDE-PFN: Prior-Data Fitted Neural PDE Solver

## Abstract

Despite recent progress in scientific machine learning (SciML), existing approaches remain impractical, as they often require explicit governing equations, impose rigid input structures, and lack generalizability across PDEs. Motivated by the success of large language models (LLMs) with broad generalizability and robustness to noisy or unreliable pre-training data, we seek to bring similar capabilities to PDE solvers. In addition, inspired by the Bayesian inference mechanisms of prior-data fitted networks (PFNs), we propose PDE-PFN, a prior-data fitted neural solver that directly approximates the posterior predictive distribution (PPD) of PDE solutions via in-context Bayesian inference. PDE-PFN builds on a PFN architecture with self- and cross-attention mechanisms of Transformer and is pre-trained on low-cost approximate solutions generated by physics-informed neural networks, serving as diverse but not necessarily exact priors. Through experiments on a range of two-dimensional PDEs, we demonstrate that PDE-PFN achieves strong generalization across heterogeneous equations, robustness under noisy priors, and zero-shot inference capability. Our approach not only outperforms task-specific baselines but also establishes a flexible and robust paradigm for advancing SciML.

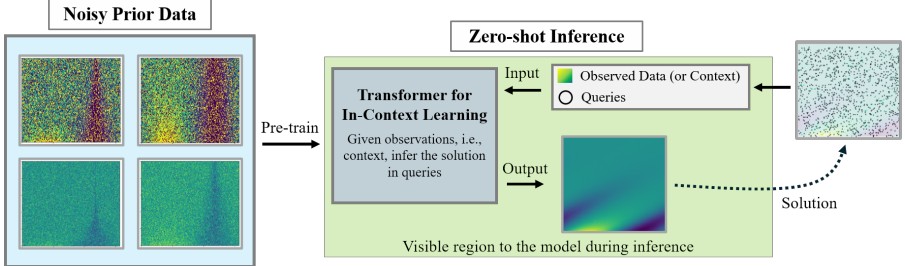

Figure 1: **The overall workflow of PDE-PFN.** Our model performs in-context learning (ICL) based on given observations (i.e., context) to infer solutions. Even when trained with an approximated PINN prior, our method obtains clean solutions due to the implicit Bayesian inference capability of ICL.

## 1 Introduction

Partial differential equations (PDEs) have long been at the core of scientific and engineering research, as they provide a mathematical description of the fundamental laws governing diverse natural and engineered systems. Numerical methods such as finite element or finite difference schemes have traditionally been employed (Quarteroni & Valli, 2008). However, these methods are not only computationally expensive in high-dimensional or multi-scale settings but also typically require complete observations of the system. With recent advances in machine learning, scientific machine learning (SciML) (Raissi et al., 2019a; Willard et al., 2022; Subramanian et al., 2023; Kim et al., 2023; 2024; Choi et al., 2024) has emerged as a promising paradigm for addressing these challenges.

Within the trend of SciML, we can see two main approaches: physics-informed neural network (PINN)-based models (Raissi et al., 2019b; Shukla et al., 2020; Yang et al., 2021; Meng et al., 2020; Yuan et al., 2022) and neural operator-based models (Li et al., 2020; Lu et al., 2021; Tran et al., 2021; Guibas et al., 2021). PINN-based models embed physical laws into the learning process by enforcing PDE residuals as constraints during training, enabling them to approximate solutions without requiring large labeled datasets. However, a critical limitation of this approach is that it presupposes explicit

knowledge of the underlying PDEs, which is often impractical in many real-world scenarios where governing equations are only partially known or entirely unavailable.

Neural operator-based models aim to learn mappings between infinite-dimensional function spaces, enabling fast inference once trained, and they have achieved notable success on benchmark PDEs. However, their applicability remains limited. First, these models are confined to the specific dataset and task they were trained on, showing poor generalization when applied to new PDEs or distributions. Second, they suffer from input rigidity: they require data to lie on fixed grids or pre-specified coordinates, which prevents flexible inference when observation points vary. These limitations collectively restrict the generalizability and flexibility of neural operator methods, underscoring the need for a more versatile framework that can transfer across diverse PDEs and input conditions.

In recent years, large language models (LLMs) have revolutionized natural language processing by introducing highly flexible and scalable architectures (Brown et al., 2020; Kaplan et al., 2020; Touvron et al., 2023; Frieder et al., 2023; Chowdhery et al., 2023). Motivated by the success of LLMs in enabling broad task generalization, scientific foundation models (SFMs) (McCabe et al., 2023; Yang et al., 2023; Yang & Osher, 2024; Hang et al., 2024) have recently been proposed. These models are typically pre-trained on a wide range of PDEs and subsequently fine-tuned for diverse downstream tasks, extending their applicability beyond the narrow scope of conventional neural operators.

Although SFMs represent a meaningful advance by enabling generalization across different PDEs, they continue to inherit other fundamental drawbacks from their neural operator backbones. In particular, they lack task-level generalization beyond the operator tasks seen during training and impose input rigidity, often constraining data to uniform grids or fixed spatial locations. To address these limitations, we draw inspiration from prior-fitted networks (PFNs) (Müller et al., 2022), which approximate the posterior predictive distribution (PPD) through Bayesian inference conditioned on prior information. Building on this idea, we propose **PDE-PFN**, a new SciML method that adopts a PFN architecture grounded in in-context learning. By directly approximating the PPD of PDE solutions, our method achieves generalizability in terms of the task and PDE type while removing structural constraints on input. In addition, its Bayesian inference enables robustness even when trained with noisy priors, which significantly reduces the data collection costs in SciML.

**Task generalizability on diverse PDEs with enhanced performance**  We first evaluated the proposed method on the convection–diffusion–reaction (CDR) equations, a standard benchmark for parameterized PDEs. Our model exhibited strong task generalizability, successfully handling diverse tasks across both parameter and spatiotemporal domains without requiring task-specific fine-tuning. In addition, although pre-trained solely on CDR equations, the model demonstrated PDE generalization by accurately predicting solutions for different PDE families with only minimal fine-tuning. Across these experiments, our method consistently matched or outperformed task-specific baselines and further surpassed existing SFMs in terms of PDE generalizability.

**Flexibility on input shape without prior physical knowledge**  Our method advances SciML by removing two key constraints that limit existing approaches. First, it predicts solutions directly from observed quantities, such as velocity and pressure, without requiring access to governing equations. This feature is particularly important in real-world applications where the underlying equations are incomplete or entirely unknown (Chien et al., 2012; Rouf et al., 2021; Beck & Kurz, 2021; Nicolaou et al., 2023; Lee & Cant, 2024). Excluding explicit equations from the input is thus an intentional design choice that broadens applicability across diverse domains. Second, in complex systems such as semiconductor manufacturing, observational data are often collected from sparse or irregularly distributed sensors rather than uniform grids (Myers & Schultz, 2000; Quirk & Serda, 2001; Chien et al., 2012). In such settings, the ability to handle inputs of arbitrary shape is essential. Our method is free from structural constraints on input representation and allows target locations to be flexibly specified as queries. Together, these properties make our approach both practical and versatile, effectively overcoming the limitations of prior SciML models. A detailed comparison with existing baselines is provided in Table 7 of Appendix D.

**Robustness to noisy prior**  For LLMs, one of the most challenging steps is collecting prior data, which typically involves crawling and cleaning sentences from the Internet. However, this process is far from perfect due to two key issues: (i) the Internet, as a data source, is inherently unreliable; and (ii) cleaning such vast amounts of data requires significant manual effort. Consequently, LLMs are often trained on incomplete or imperfect prior data. This realistic yet critical issue has been largely overlooked in the literature on SFMs, despite their similarities to LLMs.

Notably, numerical solvers are often expensive, time-consuming, and specialized for particular classes of PDEs (e.g., the finite-difference time-domain (FDTD) method for Maxwell's equations). In this work, we are the first to explore the potential of pre-training a model with PINN-based low-cost, noisy, and approximated data, demonstrating that our method can still achieve strong predictive performance under such challenging conditions.

**Zero-shot inference** Our method supports zero-shot inference for predicting PDE solutions. In contrast, ICON-LM (Yang et al., 2025) requires few-shot "demos"[1] for an unknown target operator before making predictions. Collecting such demos introduces an inherent delay, since inference cannot proceed until these examples are available (see Figure 1). By design, our approach eliminates this requirement and enables immediate inference as soon as the model is queried.

## 2 BACKGROUND

Consider a sequence of pairs $(c_1, y_1), (c_2, y_2), \cdots, (c_n, y_n)$, each within the measurable space $(\mathcal{C} \times \mathcal{Y}, \mathcal{A})$, where $c_i$ represents the spatiotemporal coordinate, $y_i$ denotes the corresponding solution, n is data size, and $\mathcal{A}$ denotes the Borel $\sigma$-algebra on the measurable space $\mathcal{C} \times \mathcal{Y}$. These pairs are drawn from a family of probability density distributions $\{p_q : q \in \mathcal{Q}\}$, where $\mathcal{Q}$ represents the *parameter space*, equipped with a $\sigma$-algebra $\mathcal{B}$, ensuring that the mappings $q \mapsto p_q(c, y)$ are measurable. The true parameter $\pi$ is an element of $\mathcal{Q}$, and the pairs $(c_i, y_i)$ are sampled according to $p_\pi$. Lacking information on $\pi$, we adopt a Bayesian framework to establish a prior distribution $\Pi$ which is defined as a probability measure on $(\mathcal{Q}, \mathcal{B})$. Then we have

$$\Pi(A \mid c, y) = \frac{\int_A p_q(c, y) \mathrm{d}\Pi(q)}{\int_\mathcal{Q} p_q(c, y) \mathrm{d}\Pi(q)},$$

for any measurable set $A \in \mathcal{B}$. This prior is updated with the observed data to form the posterior distribution, which is defined as

$$\Pi(A \mid D_n) = \frac{\int_A L_n(q) \mathrm{d}\Pi(q)}{\int_\mathcal{Q} L_n(q) \mathrm{d}\Pi(q)},$$

where $L_n(q) = \prod_{i=1}^n \frac{p_q(c_i, y_i)}{p_\pi(c_i, y_i)}$ for $A \subset \mathcal{Q}$ and $D_n = \{(c_i, y_i)\}_{i=1}^n$. The resulting posterior is

$$q_n(c, y \mid D_n) = \int_\mathcal{Q} p_q(c, y) \mathrm{d}\Pi(q \mid D_n),$$

and the posterior predictive distribution (PPD) is formulated as

$$q_n(y \mid c, D_n) = \int_\mathcal{Q} p_q(y \mid c) \, \mathrm{d}\Pi(q \mid D_n).$$

As noted by (Nagler, 2023; Walker, 2003; 2004b;a; Blasi & Walker, 2013), for a well-behaved prior, the PPD converges toward true $\pi$ as the data size $n$ increases. This aligns with findings by Blasi & Walker (2013), demonstrating that in well-specified scenarios, strong consistency is achieved as

$$\Pi^n \left(\{q : H(p_\pi, p_q) > \epsilon\}\right) \to 0 \quad \text{almost surely,} \tag{1}$$

for any $\epsilon > 0$, where $\Pi^n(A) = \int_A d\Pi(q \mid D_n)$ is the posterior measure and $H$ is the Hellinger distance defined by

$$H(p, q) = \left(\int_{\mathcal{C} \times \mathcal{Y}} (\sqrt{p} - \sqrt{q})^2\right)^{1/2}.$$

**Theorem 2.1.** *Let $D_n$, whose size is $n$, be a set of ground-truth prior data, and $\widetilde{D}_n = D_n + \eta_n$ be our (unbiased) observation, where $\eta_n$ is a zero-mean noise distribution with a finite variance. Let $p_\pi(\cdot \mid c)$ denote the true posterior, and let $\hat{p}_{\hat{\theta}}(\cdot \mid c, \widetilde{D}_n)$ be the corresponding learned (approximate) posterior for some neural network parameters $\hat{\theta}$. Suppose that the same conditions as in Lemma A.1 hold (with $\widetilde{D}$ in place of $D_n$). Then, it holds that*

$$\lim_{n \to \infty} \mathbb{E}_c \left[ H\left(\hat{p}_{\hat{\theta}}(\cdot \mid c, \widetilde{D}_n), p_\pi(\cdot \mid c)\right) \right] = 0 \quad \text{almost surely,}$$

*where $H(\cdot, \cdot)$ denotes the Hellinger distance (see Appendix A for proof).*

---

[1] In ICON and ICON-LM, a demo refers to a set of (input, output) pairs of an operator to be inferred.

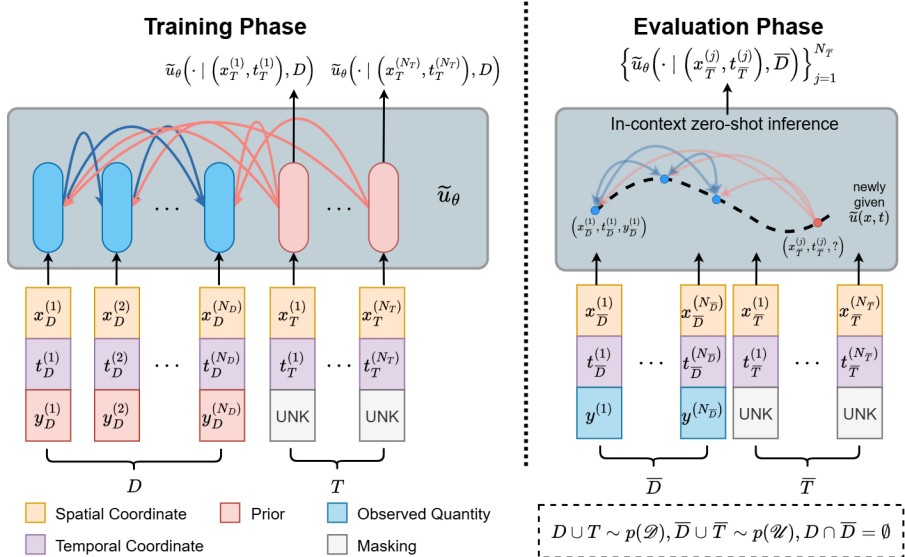

Figure 2: **A schematic diagram of our model.** *(Left)* Given the contexts $D$ and queries $T$ drawn from the prior distribution $\mathcal{D}$, our model $\tilde{u}_\theta$ is trained to infer the solutions of the queried points $T$ in the training phase. ICL is leveraged with self-attention (blue) within $D$ and cross-attention (red) from $T$ to $D$. *(Right)* In the testing phase, the pre-trained $\tilde{u}_\theta$ takes contexts $\overline{D}$ and queries $\overline{T}$ drawn from the ground truth distribution $\mathcal{U}$, and predicts for the queried points $\overline{T}$.

This result demonstrates the sensitivity of the posterior distribution approximation, accomplished by the neural network, to perturbations of the data by zero-mean noise. As the dataset size $n$ increases, the network becomes increasingly sensitive to the posterior distribution, converging to the expected value under the prior distribution. This sensitivity to the data reflects the consistency and robustness of the Bayesian inference process. Our model leverages this observation by performing Bayesian inference that incorporates (noisy) prior data, allowing it to infer robust solutions for given spatiotemporal conditions. The experimental results in Appendix H.4 confirm this behavior, showing how the network's solution converges to the true solution as $n$ increases.

## 3 METHODS

In the Background section, we denoted the spatiotemporal coordinate as $c$ for integration. However, for more clarity, as shown in Figure 2, we separate the spatiotemporal coordinate $c$ into a spatial coordinate $x$ and a temporal coordinate $t$, i.e., $c = (x, t)$. Let $\alpha$ denote a parameter vector representing the coefficients of the governing PDE dynamics. Figure 2 illustrates the schematic diagram of our model. As defined in Figure 2, $D = \{(x_D^{(i)}, t_D^{(i)}, y_D^{(i)})\}_{i=1}^{N_D}$ and $T = \{(x_T^{(i)}, t_T^{(i)}, y_T^{(i)})\}_{i=1}^{N_T}$ represent the context set and the query set for ICL, respectively. We assume these sets are independently and identically sampled from a family of probability density distributions $\{p_\alpha : \alpha \in \mathcal{Q}\}$. Here, each $y$ corresponds to the quantity at $(x, t)$ for a PDE solution $u(\alpha)$ determined by the PDE parameter $\alpha$. Depending on the problem setting, $u$ can be either the ground-truth solution or an approximated solution generated by a PINN, since we consider both exact and approximated prior data. The PPD of the solutions given the context set can be expressed as

$$\pi(y \mid x, t, D) = \int_{\mathcal{Q}} p_\alpha(y \mid x, t)\, d\Pi(\alpha \mid D).$$

This represents the posterior distribution of $y$ given $D$, capturing the most probable solution distribution for the parameter $\alpha$. In this work, we aim to predict the solution from $D$ by minimizing the mean squared error (MSE) between the PPD-derived solution and the true solution, even with noise.

**Model architecture** Since our model is essentially a prior-data fitted network (PFN), we construct its architecture by modifying certain components of the original PFN. To better handle the more complex PDE data, we first incorporate a Fourier feature embedding into the input data before it is passed to the encoder, extending the original design that was limited to tabular data. We also introduce

several modifications to the Transformer architecture. Specifically, to improve the model's ability to learn from complex data while reducing computational complexity and enhancing representational capacity, we adopt the attentive graph filter (AGF) Wi et al. (2025) structure for Transformer layers. A more detailed description of the model architecture can be found in Appendix E.

**Benchmark PDEs** We consider convection-diffusion-reaction (CDR) equations as benchmark PDEs for diverse tasks[2]:

$$u_t = \mathcal{N}(\cdot), \quad \mathcal{N}(t, x, u, \boldsymbol{\beta}, \boldsymbol{\nu}, \rho_1, \cdots, \rho_J) = -\boldsymbol{\beta} \cdot \nabla u + \nabla \cdot (\boldsymbol{\nu} \nabla u) + \sum_{j=1}^{J} \rho_j f_j(u), \quad (2)$$

where $u$ denotes the state variable; $\boldsymbol{\beta}$ is the convection (advection) velocity; $\boldsymbol{\nu}$ denotes the anisotropic diffusivity coefficient; and $\rho_1, \rho_2, \rho_3$ are scalar coefficients corresponding to the nonlinear reaction terms $f_1, f_2, f_3 : \mathbb{R} \to \mathbb{R}$, respectively. For clarity, we will refer to the two-dimensional CDR equations, including cases where certain coefficients are zero, as the family of CDR equations. This family encompasses not only generic parabolic equations but also extends to hyperbolic equations such as convection equations. Hence, the family of CDR equations contains three key terms—convection, diffusion, and reaction—each with distinct properties, making it an ideal benchmark problem. To the best of our knowledge, this work is the first to address a wide range of reaction terms within the family of CDR equations using a single unified model. We use a set of CDR-related terms and a linear combination of three nonlinear reaction terms to generate prior data. This formulation enables the incorporation of diverse reaction dynamics, making the benchmark problems more comprehensive.

**PINN-based approximations of PDE solution space** Although numerical solvers can solve the family of CDR equations, in this work, we use the predictions from PINNs as a representative example of low-cost approximated data. This allows us to evaluate the robustness of our model to noise, which is commonly encountered in real-world scenarios. From this point, we refer to these PINN-based approximations as PINN priors. To approximate the solution space for the PDEs, we construct a practical parameter subspace, $\Omega \subseteq \mathcal{Q}$, which is the collection of coefficients in Eq. (2) and has a dictionary form $\Omega = \{\boldsymbol{\alpha} := (\boldsymbol{\beta}, \boldsymbol{\nu}, \rho_1, \cdots, \rho_J)\}$. Consequently, the target exact prior $\mathcal{U}$ defined below represents the collection of solutions $u(\boldsymbol{\alpha})$ at Eq. (2) for each parameter $\boldsymbol{\alpha} \in \Omega$, where $\mathcal{X}$ and $\mathcal{T}$ correspond to the spatial and temporal domains of interest, respectively.

$$\mathcal{U} = \bigcup_{\boldsymbol{\alpha} \in \Omega} \{u(\boldsymbol{\alpha}) \,|\, u_t = \mathcal{N}(t, x, u, \boldsymbol{\alpha})\}, \quad u : \mathcal{X} \times \mathcal{T} \to \mathbb{R}.$$

Since the target exact prior data $\mathcal{U}$ is hard to obtain, we instead use a PINN prior $\mathcal{D}$ that closely approximates $\mathcal{U}$ as follows. Suppose $\tilde{u}(\boldsymbol{\alpha})$ is the prediction by PINN, which is trained to predict the PDE $u_t = \mathcal{N}(\cdot)$ (see Appendix C for details). The PINN prior $\mathcal{D}$ is a collection of approximate solutions $\tilde{u}(\boldsymbol{\alpha})$ for each $\boldsymbol{\alpha} \in \Omega$,

$$\mathcal{D} = \bigcup_{\boldsymbol{\alpha} \in \Omega} \{\tilde{u}(\boldsymbol{\alpha})\}, \quad p(\mathcal{D}) \sim p(\mathcal{U}).$$

Subsequently, the model performs the PPD of the generated prior $p(\mathcal{D})$ through ICL.

**Objective function** From a given parameter space $\Omega$, the parameter $\boldsymbol{\alpha}$ is randomly drawn i.i.d. from $\Omega$. After this, the previous $\tilde{u}(\boldsymbol{\alpha})$ is given as an input to our model $\tilde{u}_\theta$ to minimize the MSE at the predicted points, as expressed in Eq. (3). The specific MSE loss chosen to regress the solution over the spatial and temporal domain for a given $\tilde{u}(\boldsymbol{\alpha})$ is defined as

$$\mathcal{L}_{\boldsymbol{\alpha}} = \frac{1}{N_T} \sum_{j=1}^{N_T} \left[ \tilde{u}_\theta(x_T^{(j)}, t_T^{(j)} \,|\, D) - \tilde{u}(x_T^{(j)}, t_T^{(j)}) \right]^2. \quad (3)$$

In addition, we introduce a regularization term into the objective function to ensure orthogonality during the SVD process within the AGF layer (see Appendix E for detail). We denote this term as

---

[2]We note that in Eq. (2) $J$ reaction terms are considered. In previous works (Cho et al., 2023; 2024; Kang et al., 2024), only one type of reaction term is considered. Therefore, one can consider that Eq. (2) is a comprehensive dictionary (or a prior set for PFN training) of all popular CDR equations. Our experiments differ from previous works in this regard.

$\mathcal{L}_{AGF}$. To balance the trade-off between the loss and the regularization, we scale the latter by the number of context points $N_D$. Consequently, the final objective function $\mathcal{L}$ is expressed as

$$\mathcal{L} = \mathcal{L}_{\boldsymbol{\alpha}} + \frac{1}{N_D}\mathcal{L}_{AGF}.$$

The concrete flow of the training and the evaluation process is described in Appendix F and G.

## 4 EXPERIMENTS

Our experimental section is divided into three phases. In the first phase, we introduce the dataset and establish the baselines for our experiments. In the second phase, we pre-train our model on the family of 2D CDR equations and evaluate it on a variety of tasks without any additional task-specific fine-tuning. This allows us to examine the model's ability to handle diverse tasks. In the third phase, we fine-tune our model on different PDEs, including the shallow water equations (SWE) and compressible Navier-Stokes equations (CNSE), in order to assess its generalization performance across different PDEs. Table 1 summarizes the experiments in this section.

Table 1: Roadmap of the experiment section. 'Inter' and 'Extra' denote spatiotemporal interpolation and temporal extrapolation tasks, respectively. The 'Coeff/dataset' column specifies either the coefficients of the equation or the dataset used in evaluation.

| Section | Task | Coeff/Dataset | Equation | Objective |
|---------|------|---------------|----------|-----------|
| 4.2.1 | Inter & Extra | Seen | Family of 2D CDR equations | Task generalization |
| 4.2.2 | Inter & Extra | Unseen | | Task generalization in parameter space |
| 4.3.1 | Inter & Extra | Seen & Unseen | SWE | PDE generalization |
| 4.3.2 | Operator Learning | Unseen | CNSE | PDE generalization for new task |

### 4.1 EXPERIMENTAL SETUP

**Dataset** We consider a family of 2D CDR equations with three reaction terms as the pre-training dataset. This family is constructed from equations with coefficient dictionary $(\beta_x, \beta_y, \nu_x, \nu_y, \rho_1, \rho_2, \rho_3 \in \{0.0, 0.5, 1.0\})$, resulting in a total of 2,187 unique PDEs. All PDE instances are simulated from the same initial condition, which allows differences in the resulting dynamics to stem solely from the variations in coefficients. For PINN prior, each PINN is trained independently, and its predicted solutions are used as noisy priors. On average, the generated PINN priors exhibit an $L_2$ relative error of 0.02280. For PDE generalization experiments, we employ the SWE and CNSE datasets from PDEBench (Takamoto et al., 2022). These two equations are chosen because they are commonly used to validate PDE solvers in SciML. Both datasets consist of 1,000 samples generated from distinct initial conditions. The SWE dataset contains a single feature, the water height, whereas the CNSE dataset includes $x$- and $y$-velocities, pressure, and density. Detailed information on the datasets can be found in Appendix B.

**Baseline methods** We compare our model with six baseline models: deep operator network (Deep-ONet) (Lu et al., 2021), Fourier neural operator (FNO) (Li et al., 2020), factorized Fourier neural operator (F-FNO) (Tran et al., 2021), adaptive Fourier neural operator (A-FNO) (Guibas et al., 2021), latent neural operator (LNO) (Wang & Wang, 2024), Poseidon (Herde et al., 2024), and auto-regressive denoising pre-training operator Transformer (DPOT) (Hao et al., 2024). These baselines are chosen for their scalability and ability to effectively handle complex 2D PDEs. For Poseidon and DPOT, scientific foundation models with publicly available pre-trained weights, we use the released weights and perform fine-tuning. The other baselines are trained and evaluated separately on each dataset and task. For the spatiotemporal interpolation task, DeepONet is the only baseline used, excluding those that require grid inputs. The best hyperparameters for all models are selected based on their best validation performance in each task. Detailed information on the baselines used in each experiment can be found in Appendix D.

### 4.2 PRE-TRAINING ON THE FAMILY OF 2D CDR EQUATIONS AND EVALUATION

The training dataset is constructed from 6 time steps within the range $t \in [0.0, 0.5]$ at intervals of 0.1. Our model takes as an input set of contexts, which include domain coordinates paired with their solution values, along with queries, which consist of domain coordinates where the solution values are masked. We denote the version of our model pre-trained on numeric priors as **Ours** and the version pre-trained on PINN priors as **Ours (PINN)**. For the baselines, the training procedure differs

by task. In the spatiotemporal interpolation task, since DeepONet requires fixed points as branch input, we predefined the fixed points in training data by sampling and the remaining points are used as queries to be predicted. In the temporal extrapolation task, the baselines are trained to take the grid points at a time $t_n$ as input and predict the grid points at the subsequent time step $t_{n+1}$. Further information on the training procedure, such as the number of epochs, can be found in Appendix I.

### 4.2.1 Task Generalization for Seen coefficients

We first evaluate the task generalization ability in spatiotemporal interpolation and temporal extrapolation without additional task-specific fine-tuning. Both tasks are conducted within the coefficient dictionary used during pre-training, which we refer to as seen coefficients. For the spatiotemporal interpolation task, the baseline DeepONet requires fixed points as branch input. Therefore, our model is also provided with the same set of context points during evaluation to ensure a fair comparison. Specifically, we use the points at $t = 0.25$ as validation queries and the points at $t = 0.05, 0.15, 0.35$, and $0.45$ as test queries. For the temporal extrapolation task, the points at $t = 0.5$ are used as contexts, and those at $t = 0.6$ as queries for validation. At test time, the points at $t = 0.6$ serve as contexts, and those at $t = 0.7, 0.8, 0.9$, and $1.0$ are used as queries. The baselines predict a sequence of query points through rollout, whereas our model can directly predict the query points from their coordinates so that it is evaluated in this manner. The evaluation results are shown in Table 2.

Table 2: The evaluation results for the spatiotemporal interpolation and temporal extrapolation tasks applied to the family of 2D CDR equations. They are measured at the seen coefficients ($\beta_x, \beta_y, \nu_x, \nu_y, \rho_1, \rho_2, \rho_3 \in \{0.0, 0.5, 1.0\}$). The best performance is marked in **bold** and the second-best performance is marked with underline.

| Task | Metric | Ours | Ours (PINN) | DeepONet | FNO | F-FNO | A-FNO | LNO | Poseidon | DPOT |
|------|--------|------|-------------|----------|-----|-------|-------|-----|----------|------|
| Spatiotemporal interpolation | $L_1$ Abs | **0.01218** | 0.01507 | 0.01922 | - | - | - | - | - | - |
| | $L_2$ Rel | **0.01643** | 0.02057 | 0.02479 | | | | | | |
| | $L_\infty$ Rel | **0.04600** | 0.06584 | 0.05980 | | | | | | |
| Temporal extrapolation | $L_1$ Abs | **0.01474** | 0.01940 | 0.06529 | 0.03302 | 0.02731 | 0.05800 | 0.02876 | 0.14252 | 0.04924 |
| | $L_2$ Rel | **0.02261** | 0.02705 | 0.08212 | 0.04058 | 0.03430 | 0.07915 | 0.03665 | 0.15580 | 0.08587 |
| | $L_\infty$ Rel | **0.08444** | 0.08802 | 0.22802 | 0.09553 | 0.09877 | 0.28905 | 0.10948 | 0.28285 | 0.08587 |

The evaluation results demonstrate that both versions of our model outperform the baselines on both tasks. This indicates that once trained on a single dataset, our model can achieve strong performance without requiring task-specific fine-tuning. Even when trained with noisy PINN priors, our model surpasses the baselines in two metrics, confirming its robustness to noise in the priors. Additionally, we conduct experiments where the PINN prior is provided as a context set during evaluation in Appendix H.1.

### 4.2.2 Task Generalization for Unseen coefficients

We further evaluate the task generalization ability on both tasks in parameter space using the family of 2D CDR equations with unseen coefficients, without any additional fine-tuning. The unseen coefficients are categorized into two cases: coefficient interpolation and coefficient extrapolation. Coefficient interpolation experiments are performed on the equations with intermediate coefficient values not seen during pre-training, ($\beta_x, \beta_y, \nu_x, \nu_y, \rho_1, \rho_2, \rho_3 \in \{0.25, 0.75\}$). Coefficient extrapolation experiments, on the other hand, are conducted on the equations with larger coefficient values than those used in pre-training, ($\beta_x, \beta_y, \nu_x, \nu_y, \rho_1, \rho_2, \rho_3 \in \{1.25, 1.5\}$). For evaluation with unseen coefficients in the temporal extrapolation task, we modify the temporal extrapolation setup to remain within the temporal domain used during training: the points at $t = 0.1$ are provided as contexts, and models are asked to predict the values at $t = 0.2, 0.3, 0.4$, and $0.5$. The evaluation results are summarized in Table 3.

The evaluation results show that both versions of our model outperform the baselines on both tasks under coefficient interpolation and extrapolation. This shows that our models can make accurate predictions not only on PDEs with coefficients seen during training but also on PDEs with unseen coefficients, without requiring additional fine-tuning. In other words, the models are able to generalize across the parameter space, achieving both interpolation and extrapolation.

### 4.3 Fine-tuning on different PDEs and Evaluation

To evaluate the PDE generalization ability of our model, we conduct experiments on heterogeneous PDEs beyond the family of 2D CDR equations. On the shallow water equations (SWE) dataset, we

Table 3: The evaluation results for the spatiotemporal interpolation and temporal extrapolation tasks applied to the family of 2D CDR equations. They are measured at the unseen coefficients. The best performance is marked in **bold** and the second-best performance is marked with underline.

| Coefficients | Task | Metric | Ours | Ours (PINN) | DeepONet | FNO | F-FNO | A-FNO | LNO | Poseidon | DPOT |
|---|---|---|---|---|---|---|---|---|---|---|---|
| Interpolation | Spatiotemporal interpolation | $L_1$ Abs | **0.01241** | 0.01277 | 0.01347 | - | - | - | - | - | - |
| | | $L_2$ Rel | **0.01658** | 0.01761 | 0.01731 | | | | | | |
| | | $L_\infty$ Rel | 0.04274 | 0.05503 | **0.04114** | | | | | | |
| | Temporal extrapolation | $L_1$ Abs | 0.08957 | **0.08709** | 0.11918 | 0.11625 | 0.11194 | 0.12220 | 0.11085 | 0.17740 | 0.13879 |
| | | $L_2$ Rel | **0.11990** | 0.12465 | 0.14903 | 0.14658 | 0.14348 | 0.18191 | 0.14913 | 0.21340 | 0.15085 |
| | | $L_\infty$ Rel | 0.29526 | 0.36472 | 0.35196 | **0.28689** | 0.30813 | 0.64429 | 0.33280 | 0.39890 | 0.15085 |
| Extrapolation | Spatiotemporal interpolation | $L_1$ Abs | 0.01661 | **0.01582** | 0.03576 | - | - | - | - | - | - |
| | | $L_2$ Rel | 0.02398 | **0.02038** | 0.04669 | | | | | | |
| | | $L_\infty$ Rel | 0.08324 | **0.06890** | 0.10392 | | | | | | |
| | Temporal extrapolation | $L_1$ Abs | **0.01027** | 0.02063 | 0.05514 | 0.02652 | 0.02854 | 0.05013 | 0.05076 | 0.14661 | 0.05186 |
| | | $L_2$ Rel | **0.01068** | 0.02169 | 0.06450 | 0.03043 | 0.03300 | 0.06351 | 0.06347 | 0.15650 | 0.05189 |
| | | $L_\infty$ Rel | **0.01661** | 0.03474 | 0.16555 | 0.06124 | 0.08343 | 0.20955 | 0.15091 | 0.27502 | 0.05189 |

assessed whether our model could generalize, via fine-tuning, to heterogeneous PDEs on the same tasks as in the family, namely spatiotemporal interpolation and temporal extrapolation. The baselines for each task are trained in the same manner as in the family. On the CNSE dataset, we further examined whether the model could generalize to a new task, operator learning, under a different PDE.

### 4.3.1 PDE GENERALIZATION FOR SHALLOW WATER EQUATIONS

To evaluate the PDE generalization ability of our model, we conduct experiments on the SWE, assessing whether the tasks defined in the family of 2D CDR equations can be successfully transferred to this heterogeneous PDE by minimal fine-tuning. Out of the total 1,000 samples, 700 are used for training, and the training, validation, and test datasets are constructed over the time interval in the same manner as for the family of 2D CDR equations. The test dataset generated from the initial condition samples used during training is referred to as the seen dataset, while the test dataset generated from the remaining 300 samples is referred to as the unseen dataset. For the baselines, training for each task is carried out in the same manner as in pre-training.

Table 4: The evaluation results for the spatiotemporal interpolation and temporal extrapolation tasks applied to SWE. They are measured at the seen/unseen datasets. The best performance is marked in **bold** and the second-best performance is marked with underline.

| Evaluation dataset | Task | Metric | Ours | Ours (PINN) | DeepONet | FNO | F-FNO | A-FNO | LNO | Poseidon | DPOT |
|---|---|---|---|---|---|---|---|---|---|---|---|
| Seen | Spatiotemporal interpolation | $L_1$ Abs | **0.00208** | 0.00269 | 0.00271 | - | - | - | - | - | - |
| | | $L_2$ Rel | **0.00739** | 0.00822 | 0.00997 | | | | | | |
| | | $L_\infty$ Rel | **0.05897** | 0.06950 | 0.06951 | | | | | | |
| | Temporal extrapolation | $L_1$ Abs | 0.01548 | **0.01418** | 0.06994 | 0.03271 | 0.02292 | 0.05036 | 0.02902 | 0.02964 | 0.03327 |
| | | $L_2$ Rel | 0.03273 | **0.03221** | 0.08319 | 0.05429 | 0.04110 | 0.07436 | 0.05269 | 0.05208 | 0.05860 |
| | | $L_\infty$ Rel | **0.20590** | 0.24596 | 0.28736 | 0.24954 | 0.24478 | 0.38030 | 0.24832 | 0.25668 | 0.29727 |
| Unseen | Spatiotemporal interpolation | $L_1$ Abs | **0.00213** | 0.00272 | 0.00278 | - | - | - | - | - | - |
| | | $L_2$ Rel | **0.00754** | 0.00828 | 0.01008 | | | | | | |
| | | $L_\infty$ Rel | **0.05986** | 0.06981 | 0.07047 | | | | | | |
| | Temporal extrapolation | $L_1$ Abs | 0.01571 | **0.01448** | 0.07047 | 0.03342 | 0.02371 | 0.05099 | 0.02877 | 0.03021 | 0.03372 |
| | | $L_2$ Rel | 0.03323 | **0.03288** | 0.08347 | 0.05530 | 0.04218 | 0.07513 | 0.05225 | 0.05306 | 0.05914 |
| | | $L_\infty$ Rel | **0.20686** | 0.24686 | 0.28441 | 0.24980 | 0.24378 | 0.38196 | 0.24788 | 0.25710 | 0.29757 |

The experimental results in Table 4 show that both versions of our model outperform the baselines on both spatiotemporal interpolation and temporal extrapolation tasks, regardless of whether the dataset is seen or unseen. Specifically, Ours achieves the best performance on spatiotemporal interpolation, while Ours (PINN) performs best on temporal extrapolation. These results demonstrate that our approach can generalize the two tasks verified in pre-training to different PDEs through fine-tuning, while also retaining generalization ability to unseen datasets.

### 4.3.2 PDE GENERALIZATION FOR COMPRESSIBLE NAVIER-STOKES EQUATIONS

To further examine the PDE generalization ability of our model, we conduct experiments on the compressible Navier–Stokes equations (CNSE), focusing in particular on learning the time trajectory predicting operator. This experiment is designed not only to verify generalization across heterogeneous PDEs but also to assess the potential of our approach in learning an operator that was not included in previous experiments. The dataset consists of 1,000 samples, of which 700 are used for training, 100 for validation, and 200 for testing. The target operator is defined as follows: given the initial condition of four features, the models are required to predict the solution of them at later times

$t = 0.25, 0.5, 0.75$, and $1.0$ (see Appendix B.3 for a formal description). Baseline models except DeepONet and DPOT are trained and evaluated using rollout predictions. In contrast, DeepONet and our model directly predict the solutions given the initial condition. Additionally, since DPOT allows the output timestep length to be adjusted, it also predicts the solutions in a single step.

Table 5: The evaluation results for the operator learning task applied to CNSE. The best performance is marked in **bold** and the second-best performance is marked with underline.

| Task | Metric | Ours | Ours (PINN) | DeepONet | FNO | F-FNO | A-FNO | LNO | Poseidon | DPOT |
|------|--------|------|-------------|----------|-----|-------|-------|-----|----------|------|
| Operator learning | $L_1$ Abs | 0.09497 | 0.09429 | 0.17797 | 0.09581 | **0.08734** | 0.13637 | 0.10366 | 0.13313 | 0.09164 |
| | $L_2$ Rel | 0.54833 | **0.54369** | 1.09567 | 0.57905 | 0.55420 | 0.65798 | 0.62536 | 0.66904 | 0.67869 |
| | $L_\infty$ Rel | 0.63533 | **0.63374** | 1.09165 | 0.70708 | 0.72876 | 0.85729 | 0.66936 | 0.92909 | 1.15028 |

The experimental results in Table 5 show that both versions of our model outperform the baselines on two relative error metrics. Notably, although DPOT's pre-training corpus already includes the CNSE dataset from PDEBench, our models still achieve superior performance over DPOT on both metrics. While DPOT was originally evaluated on predicting one time step ahead, in our setting the task was extended to predicting up to four time steps ahead. As a result, DPOT exhibited lower performance than reported in the original paper. Experimental results confirm the generalization capability of our approach for operator learning on heterogeneous PDEs involving multiple variables.

To further verify PDE generalization and input flexibility, we conduct supplementary experiments. First, we evaluate operator learning in the same form as in the CNSE experiments on the Airfoil dataset, which is constructed with an irregular mesh structure. In addition, we evaluate operator learning across different features, rather than within the same features, using the Darcy Flow dataset. Detailed information on these experiments is provided in Appendix H.

## 5 RELATED WORKS

**In-context learning** Transformers have shown remarkable ICL abilities in various studies. They generalize to unseen tasks by emulating Bayesian predictors (Panwar et al., 2024) and linear models (Zhang et al., 2024), while also efficiently performing Bayesian inference through Prior-Data Fitted Networks (PFNs) (Müller et al., 2021). Their robustness extends to learning different function classes, such as linear and sparse linear functions, decision trees, and two-layer neural networks, even under distribution shifts (Garg et al., 2022). Furthermore, Transformers can adaptively select algorithms based on input sequences, achieving near-optimal performance on tasks like noisy linear models (Bai et al., 2023). They are also fast and effective for tabular data classification (Hollmann et al., 2022).

**Scientific foundation models** Recent studies have advanced in-context operator learning and PDE solving through Transformer-based models. The work Ye et al. (2024) introduces PDEformer, a versatile model for solving 1D PDEs with high accuracy in inverse problems. In-context operator learning has also been extended to multi-modal frameworks, as seen in Yang et al. (2025), where ICON-LM integrates natural language with equations to outperform traditional models. Additionally, Yang & Osher (2024) and Yang et al. (2023) demonstrate the generalization capabilities of ICON in solving various PDE-related tasks, highlighting ICON's few-shot learning performance across various problems in differential equations. Several other studies have addressed the problem of solving diverse PDEs using a single trained model (Hang et al., 2024; McCabe et al., 2023; Herde et al., 2024; Hao et al., 2024). However, many of these approaches rely on symbolic PDE information, true or near-true solutions, and/or do not support zero-shot, in-context learning, making their objectives different from ours.

## 6 CONCLUSION

We presented PDE-PFN, a new method for scientific machine learning that integrates in-context learning and Bayesian inference to directly approximate the posterior predictive distribution of PDE solutions. Our experiments demonstrated that PDE-PFN achieves both task and PDE generalization, handles flexible input structures without relying on governing equations, and remains robust even under noisy priors, while also enabling zero-shot inference. Together, these properties establish PDE-PFN as a flexible and robust foundation for advancing scientific machine learning. Since our current study focuses on demonstrating the feasibility of a PFN in SciML, its empirical verification is limited to two-dimensional PDEs and has not yet been validated on higher-dimensional or more complex systems. Future work will extend PDE-PFN to such challenging settings, further enhancing its applicability to real-world scientific and engineering problems.

## ETHICS STATEMENT

This research adheres to the ethical standards required for scientific inquiry. We have considered the potential societal impacts of our work and have found no clear negative implications. We also see no direct path from our model to malicious uses such as surveillance, disinformation, or privacy breaches. On the other hand, we recognize that improvements in PDE-solving capabilities could eventually be applied in real-world applications, such as climate modeling, biomedical simulations, or engineering systems.

All experiments were conducted in compliance with relevant laws and ethical guidelines, ensuring the integrity of our findings. We are committed to transparency and reproducibility in our research processes.

## REPRODUCIBILITY

We are committed to ensuring the reproducibility of our research. All experimental procedures, data sources, and algorithms used in this study are clearly documented in the paper. The code will be provided as the supplementary material and be made publicly available upon publication, allowing others to validate our findings and build upon our work.

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

# A THE PROOF OF THEOREM 2.1

In the following lemma, we first prove for $D_n$, i.e., noise-free. We then extend it to the case where the noisy data $\tilde{D}_n$ is used for training.

**Lemma A.1.** *Suppose that for any $\epsilon > 0$, there exists a Transformer parameterized by $\hat{\theta}$ such that*

$$\mathbb{E}_c \left[ KL \left( p_{\hat{\theta}}(\cdot \mid c, D_n), q_n(\cdot \mid c, D_n) \right) \right] < \epsilon,$$

*for any realization of $D_n$. If the posterior consistency condition Eq. (1) holds, and for any $q \in \mathcal{Q}$, $p_q(c) = p_\pi(c)$ almost everywhere on $\mathcal{X}$, then the following holds almost surely:*

$$\mathbb{E}_c \left[ H \left( p_{\hat{\theta}}(\cdot \mid c, D_n), p_\pi(\cdot \mid c) \right) \right] \xrightarrow{n \to \infty} 0.$$

*Proof.* For any two probability distributions $p$ and $q$, recall that the Hellinger distance satisfies

$$H(p,q)^2 = \frac{1}{2} \int \left( \sqrt{p(\theta)} - \sqrt{q(\theta)} \right)^2 d\theta \leq \frac{1}{2} KL(p\|q).$$

Thus, if we can show that the Kullback–Leibler divergence $KL\left(\hat{p}_{\hat{\theta}}(\cdot \mid c, \tilde{D}) \,\|\, p_\pi(\cdot \mid c)\right)$ vanishes asymptotically, then the convergence in Hellinger distance follows. For any $n, \epsilon$, we derive that

$$\mathbb{E}_c \left[ H \left( p_{\hat{\theta}}(\cdot \mid c, D_n), p_\pi(\cdot \mid c) \right) \right] \overset{(1)}{\leq} \mathbb{E}_c \left[ H \left( p_{\hat{\theta}}(\cdot \mid c, D_n), q_n(\cdot \mid c, D_n) \right) \right]$$
$$+ \mathbb{E}_c \left[ H \left( q_n(\cdot \mid c, D_n), p_\pi(\cdot \mid c) \right) \right]$$
$$\overset{(2)}{\leq} \sqrt{\frac{1}{2} \mathbb{E}_c \left[ KL \left( p_{\hat{\theta}}(\cdot \mid c, D_n), q_n(\cdot \mid c, D_n) \right) \right]}$$
$$+ \mathbb{E}_c \left[ H \left( q_n(\cdot \mid c, D_n), p_\pi(\cdot \mid c) \right) \right]$$
$$\overset{(3)}{\leq} \sqrt{\frac{\epsilon}{2}} + \mathbb{E}_c \left[ 1 - \int_{\mathcal{Y}} \sqrt{\int p_q(y \mid c) p_\pi(y \mid c) \mathrm{d}\Pi^n(q)} \mathrm{d}y \right]^{1/2}$$
$$\leq \sqrt{\frac{\epsilon}{2}} + \left[ 1 - \int_{\mathcal{C}} p_\pi(c) \int_{\mathcal{Y}} \frac{1}{p_\pi(c)} \sqrt{\int q(y,c) p_\pi(y,c) \mathrm{d}\Pi^n(q)} \mathrm{d}y \mathrm{d}c \right]^{1/2}$$
$$\leq \sqrt{\frac{\epsilon}{2}} + \left[ 1 - \int_{\mathcal{C}} \int_{\mathcal{Y}} \int \sqrt{p_q(y,c) p_\pi(y,c)} \mathrm{d}\Pi^n(q) \mathrm{d}y \mathrm{d}c \right]^{1/2}$$
$$= \sqrt{\frac{\epsilon}{2}} + \left[ \int H \left( p_q, p_\pi \right)^2 \mathrm{d}\Pi^n(q) \right]^{1/2}$$
$$\leq \sqrt{\frac{\epsilon}{2}} + \left[ \int H \left( p_q, p_\pi \right) d\Pi^n(q) \right]^{1/2}$$
$$\overset{(4)}{=} \sqrt{\frac{\epsilon}{2}} + \left[ \int_{\{q : H(p_\pi, p_q) > \epsilon\}} H \left( p_q, p_\pi \right) \mathrm{d}\Pi^n(q) \right]^{1/2}$$
$$+ \left[ \int_{\{q : H(p_\pi, p_q) \leq \epsilon\}} H \left( p_q, p_\pi \right) \mathrm{d}\Pi^n(q) \right]^{1/2}$$
$$\overset{(5)}{=} \sqrt{\frac{\epsilon}{2}} + (\Pi^n(\{q : H(p_\pi, p_q) > \epsilon\}) + \epsilon)^{1/2} \to \sqrt{\frac{\epsilon}{2}} + \sqrt{\epsilon} \quad \text{a.s.}$$

The first inequality ($\overset{(1)}{\leq}$) is derived from the triangle inequality for the Hellinger distance, which states that for any intermediate distribution $q(\cdot \mid c, D_n)$, we have

$$H \left( p_{\hat{\theta}}(\cdot \mid c, D_n), p_\pi(\cdot \mid c) \right) \leq H \left( p_{\hat{\theta}}(\cdot \mid c, D_n), q_n(\cdot \mid c, D_n) \right) + H \left( q_n(\cdot \mid c, D_n), p_\pi(\cdot \mid c) \right).$$

The second inequality ($\overset{(2)}{\le}$) uses the fact that the Hellinger distance $H(p, q)$ is bounded above by the square root of the KL divergence $KL(p \parallel q)$, such that

$$H(p,q)^2 \le \frac{1}{2} KL(p \parallel q).$$

Thus, we can bound the Hellinger distance by the KL divergence. In the third inequality ($\overset{(3)}{\le}$), we make use of the assumption

$$\mathbb{E}_c \left[ KL \left( p_{\hat{\theta}}(\cdot \mid c, D_n), q_n(\cdot \mid c, D_n) \right) \right] < \epsilon,$$

and utilize the definition of the Hellinger distance. In ($\overset{(4)}{=}$), we partition the domain into two regions– one where the Hellinger distance $H(p_\pi, p_q)$ exceeds $\epsilon$ and another where it is less than or equal to $\epsilon$–and use this partitioning to demonstrate the inequality.

Finally, in ($\overset{(5)}{=}$), by posterior consistency, the region where the Hellinger distance is greater than $\epsilon$ vanishes as $n \to \infty$ such that

$$\Pi^n \left\{ q : H(p_\pi, p_q) > \epsilon \right\} \to 0 \quad \text{almost surely.}$$

Since $\epsilon$ is arbitrary, we can conclude that

$$\mathbb{E}_c \left[ H \left( p_{\hat{\theta}}(\cdot \mid c, D_n), p_\pi(\cdot \mid c) \right) \right] \xrightarrow{n \to \infty} 0 \quad \text{almost surely.}$$

$\square$

Based on Lemma A.1, we can prove Theorem 2.1 as follows.

**Theorem A.2.** *Let $D_n$ be a set of ground-truth prior data, whose size is $n$, and $\widetilde{D}_n = D_n + \eta_n$, where $\eta_n$ is a zero-mean noise distribution with a finite variance, be our observation. Therefore, $\widetilde{D}_n$ is an unbiased observation of $D_n$. Let $p_\pi(\cdot \mid c)$ denote the true posterior, and let $\hat{p}_{\hat{\theta}}(\cdot \mid c, \widetilde{D}_n)$ be the corresponding learned (approximate) posterior for some neural network parameter $\hat{\theta}$. Suppose that the same conditions as in Lemma A.1 hold (with $\widetilde{D}$ in place of $D_n$). Then, for any $\epsilon > 0$, it holds that*

$$\lim_{n \to \infty} \mathbb{E}_c \left[ H \left( \hat{p}_{\hat{\theta}}(\cdot \mid c, \widetilde{D}_n), \, p_\pi(\cdot \mid c) \right) \right] = 0 \quad \text{almost surely,}$$

*where, $H(\cdot, \cdot)$ denotes the Hellinger distance.*

*Proof.* In the noise-free setting of the previous lemma, it is established that

$$\mathbb{E}_c \left[ H \left( \hat{p}_{\hat{\theta}}(\cdot \mid c, D_n), \, p_\pi(\cdot \mid c) \right) \right] \to 0 \quad \text{as } n \to \infty \quad \text{(a.s.).}$$

In our case, since $\widetilde{D}_n = D_n + \eta_n$ with $\mathbb{E}[\eta_n] = 0$, every occurrence of $D_n$ is replaced by $\widetilde{D}$, and the additional noise introduces an extra expectation $\mathbb{E}_{\eta_n}[\cdot]$ in the relevant integrals. More precisely, one has

$$KL \left( \hat{p}_{\hat{\theta}}(\cdot \mid c, \widetilde{D}) \parallel q_n(\cdot \mid c, \widetilde{D}) \right) = KL \left( \hat{p}_{\hat{\theta}}(\cdot \mid c, D_n + \eta_n) \parallel q_n(\cdot \mid c, D_n + \eta_n) \right).$$

Because $\eta_n$ is unbiased and of finite variance, the extra terms arising from the noise remain bounded and can be integrated out without affecting the asymptotic convergence rate established in the noise-free case. By posterior consistency, the region where $H \left( p_\pi(\cdot), p_q(\cdot) \right) > \epsilon$ has vanishing measure as $n \to \infty$, i.e. $\Pi \left\{ q : H(p_\pi, p_q) > \epsilon \right\} \longrightarrow 0$ almost surely. Uniform integrability and continuity arguments used to control the posterior concentration remain valid, as the additional terms contributed by $\eta_n$ are absorbed by the outer expectation $\mathbb{E}_{\eta_n}[\cdot]$. Thus, one concludes that

$$\lim_{n \to \infty} \mathbb{E}_c \left[ H \left( \hat{p}_{\hat{\theta}}(\cdot \mid c, \widetilde{D}), \, p_\pi(\cdot \mid c) \right) \right] = 0 \quad \text{almost surely.}$$

This completes the proof that the consistency result of Lemma A.1 extends to the case where the prior data is perturbed by small unbiased noise. $\square$

# B  DATASETS

## B.1  FAMILY OF 2D CONVECTION-DIFFUSION-REACTION EQUATIONS

The formulation of the family of 2D CDR equations with three reaction terms, used in Section 4.2, is given as follows.

$$\text{2D CDR: } u_t + \boldsymbol{\beta} \cdot \nabla u - \nabla \cdot (\boldsymbol{\nu} \nabla u) - \sum_{j=1}^{3} \rho_j f_j(u) = 0, \ x \in \mathcal{X}, \ t \in \mathcal{T},$$

where $\mathcal{X} = [0, \pi] \times [0, \pi]$ and $\mathcal{T} = [0, 1]$ are solution domain with a resolution of $32 \times 32 \times 11$. For the family of 2D CDR equations, we use analytic solutions whenever they are available. When an analytic solution is not available, the solution is generated using the spectral method, following the same approach as in (Cho et al., 2023). The solutions are computed over the time interval $t \in [0.0, 1.0]$ with 1001 time steps at intervals of 0.001, from which 21 time steps at intervals of 0.05 (from $t = 0.0$ to $t = 1.0$) are used in the experiments. We pre-train our model on the family of 2D CDR equations with coefficients $(\beta_x, \beta_y, \nu_x, \nu_y, \rho_1, \rho_2, \rho_3 \in 0.0, 0.5, 1.0)$. This results in a total of 2,187 unique PDEs. For PINN prior, because a PINN can be trained on only a single PDE at a time, we train an individual PINN for each of the 2,187 distinct 2D CDR equations.

## B.2  SHALLOW WATER EQUATION

The 2D shallow water equation (SWE) dataset used in Section 4.3.1 is obtained from PDEBench (Takamoto et al., 2022)[3]. The SWE is widely used as a benchmark dataset and corresponds to a hyperbolic PDE of the following form:

$$\partial_t h + \partial_x (hu) + \partial_y (hv) = 0,$$

$$\partial_t (hu) + \partial_x \left( u^2 h + \frac{1}{2} g_r h^2 \right) + \partial_y (uvh) = -g_r h \partial_x b,$$

$$\partial_t (hv) + \partial_y \left( v^2 h + \frac{1}{2} g_r h^2 \right) + \partial_x (uvh) = -g_r h \partial_y b,$$

where $u$ and $v$ denote the horizontal and vertical velocities, $h$ is the water depth, and $g_r$ is the gravitational acceleration. The SWE is derived from the general Navier–Stokes equations and is well suited for modeling free-surface flow problems.

The dataset contains simulation results for the water height $h$ and is generated using the Py-Claw (Ketcheson et al., 2012) Python package. It consists of samples produced from 1,000 different initial conditions. The simulation domain is $\mathcal{X} = [-2.5, 2.5]^2$ over the time interval $\mathcal{T} = [0.0, 1.0]$, with a resolution of $128 \times 128 \times 101$. From this, we subsample 21 time steps at intervals of 0.05 for use in our experiments.

## B.3  2D COMPRESSIBLE NAVIER-STOKES EQUATION

The 2D compressible Navier–Stokes equations dataset, referred to as CNSE, used in Section 4.3.2 was obtained from PDEBench. The Navier–Stokes equation is one of the most widely adopted benchmarks for evaluating SciML PDE solvers, and in this work we focus on the compressible fluid case. The governing equations are given by

$$\partial_t \rho + \nabla \cdot (\rho \mathbf{v}) = 0,$$

$$\rho \left( \partial_t \mathbf{v} + \mathbf{v} \cdot \nabla \mathbf{v} \right) = -\nabla p + \eta \Delta \mathbf{v} + \left( \zeta + \frac{\eta}{3} \right) \nabla (\nabla \cdot \mathbf{v}),$$

$$\partial_t \left( \epsilon + \frac{1}{2} \rho v^2 \right) + \nabla \cdot \left[ \left( \epsilon + p + \frac{1}{2} \rho v^2 \right) \mathbf{v} - \mathbf{v} \cdot \boldsymbol{\sigma}' \right] = 0,$$

where $\rho$ denotes the mass density, $\mathbf{v}$ the velocity, $p$ the gas pressure, $\epsilon = p/(\Gamma - 1)$ the internal energy with $\Gamma = 5/3$, $\boldsymbol{\sigma}'$ the viscous stress tensor, and $\eta, \zeta$ the shear and bulk viscosity, respectively.

---

[3]Dataset source: https://github.com/pdebench/PDEBench

Numerical solutions were computed in the spatial domain $\mathcal{X} = [0.0, 1.0]^2$ and temporal domain $\mathcal{T} = [0.0, 1.0]$ using a second-order HLLC scheme (Toro et al., 1994) with the MUSCL method (Van Leer, 1979) for the inviscid part, and a central difference scheme for the viscous part. We use the solution computed by setting the Mach number as $M = |\mathbf{v}|/c_s = 1.0$, where $c_s = \Gamma p/\rho$ is the sound velocity, and both viscosity coefficients as $\eta = \zeta = 0.1$. Periodic boundary conditions were applied, and distinct initial conditions were generated from random fields. The dataset contains simulation results for $x$- and $y$-velocities $\mathbf{v}$, pressure $p$, and density $\rho$. The dataset consists of 1,000 samples simulated from different initial conditions, originally computed on a $128 \times 128 \times 21$ grid. Each feature in the dataset was normalized individually for the experiment. For our experiments, we employed a subsampled version of the dataset with resolution $64 \times 64 \times 5$. In the experiment we consider a field $\mathbf{u}(x, t) = [\mathbf{v}(x, t), p(x, t), \rho(x, t)] \in \mathbb{R}^4$ for $x \in \mathcal{X}$, $t \in \mathcal{T}$. Here $\mathbf{u}(x, t)$ denotes a four-dimensional vector field representing four features of the system and the initial condition $\mathbf{u}(x, 0.0) = \mathbf{u}_0(x)$ for $x \in \mathcal{X}$. We are interested in learning the trajectory-predicting operator $\mathcal{G}^\dagger : \mathbf{u}_0(x) \mapsto (\mathbf{u}(x, 0.25), \mathbf{u}(x, 0.5), \mathbf{u}(x, 0.75), \mathbf{u}(x, 1.0))$, which maps the initial state $\mathbf{u}_0 \in L^2((0.0, 1.0)^2; \mathbb{R}^4)$ to the solution at four future time instances.

### B.4  AIRFOIL

The Airfoil dataset used in Section H.2 was obtained from the dataset provided by (Li et al., 2023)[4]. This dataset was originally generated in (Pfaff et al., 2020) by solving the Euler equations for compressible flow using the finite volume method built into the SU2 library (Palacios et al., 2013)[5]. The Euler equations have the following formulation:

$$\partial_t \rho + \nabla \cdot (\rho \mathbf{v}) = f_1,$$
$$\partial_t(\rho \mathbf{v}) + \nabla \cdot (\rho \mathbf{v} \otimes \mathbf{v} + p\mathbb{I}) = \mathbf{f}_2,$$
$$\partial_t(\rho E) + \nabla \cdot (\rho E \mathbf{v} + p\mathbf{v}) = f_3,$$
$$\rho := \rho(x, t), \ \mathbf{v} := \mathbf{v}(x, t), \ p := p(x, t), x \in \mathcal{X}, \ t \in [0, T],$$

where $\rho$ denotes the density, $\mathbf{v}$ the velocity field, $p$ the pressure, $E$ the total energy per unit mass, and $f_1, \mathbf{f}_2, f_3$ are generic source terms. Li et al. (2023) formulated this problem as a non-Markovian initial value problem and used a significantly larger time step size. The dataset spans $t \in \mathcal{T} = [0.0, 4.8]$ with 101 time steps, each spaced by a time interval of 0.24. At each timepoint, the dataset contains values for 5,233 irregular mesh nodes, including node position, node type, velocity, pressure, and density. Similar to CNSE, we consider a field $\mathbf{u}(x, t) = [\mathbf{v}(x, t), p(x, t), \rho(x, t)] \in \mathbb{R}^4$ for $x \in \mathcal{X}$, $t \in \mathcal{T}$. Here $\mathbf{u}(x, t)$ denotes a four-dimensional vector field representing four features of the system. The initial condition is given by $\mathbf{u}(x, 0) = \mathbf{u}_0(x)$ for $x \in \Omega$. We want to learn the trajectory-predicting operator $\mathcal{G}^\dagger : \mathbf{u}_0(x) \mapsto (\mathbf{u}(x, 1.2), \mathbf{u}(x, 2.4), \mathbf{u}(x, 3.6), \mathbf{u}(x, 4.8))$, which maps the initial state $\mathbf{u}_0 \in L^2(\mathcal{X}; \mathbb{R}^4)$ to the solution at four future time instances.

### B.5  DARCY FLOW

The Darcy Flow dataset used in Section H.3 is also obtained from PDEBench. Darcy Flow is a widely adopted benchmark for evaluating operator learning models, formulated as a time-independent elliptic PDE. The governing equation is given by:

$$-\nabla(a(x)\nabla u(x)) = f(x), \quad x \in \mathcal{X},$$
$$u(x) = 0, \quad x \in \partial\mathcal{X},$$

where the source term $f(x)$ is fixed to a constant value of $0.1$. The dataset consists of steady-state solutions $u$ corresponding to the viscosity fields $a$. To obtain these solutions, we simulate the following time-dependent formulation with random initial conditions until convergence to steady state:

$$\partial_t u(x, t) - \nabla(a(x)\nabla u(x, t)) = f(x), \quad x \in \mathcal{X}$$

Numerical solutions are simulated using a second-order central difference scheme in both time and space. Each data sample is represented as an input–output pair, where the input is a viscosity field

---

[4]Dataset Source: https://github.com/BaratiLab/OFormer
[5]Dataset and numerical solver source: https://github.com/merantix-momentum/gnn-bvp-solver

$a(x)$ and the output is the corresponding steady-state solution $u(x)$. Both inputs and outputs are discretized on a $128 \times 128$ spatial grid. The dataset contains 2,000 samples, each generated from distinct random initial conditions within the simulation domain $\mathcal{X} = [0.0, 1.0]^2$. Unlike the CNSE setting, here we aim to learn the operator mapping a viscosity field $a(x) \in L^2\big((0.0, 1.0)^2; \mathbb{R}\big)$ to the corresponding steady-state solution $u(x)$ for $x \in \mathcal{X}$. Specifically, we want to learn the operator $\mathcal{G}^\dagger : a(x) \mapsto u(x)$.

The specific information about PDE types used in each experiment is described in Table 6. Moreover, the detailed values regarding how many training, validation, and testing data points in each experiment are provided in Appendix I. For the spatiotemporal interpolation with unseen coefficients in the family of 2D CDR equation experiments, we use the same time range as in the corresponding task with seen coefficients.

Table 6: Time range of data points used for training, validation, and testing in all experiments.

| PDE (Section) | Task | Train | Valid | | Test | |
|---|---|---|---|---|---|---|
| | | | Context | Query | Context | Query |
| Family of 2D CDR equations (Section 4.2) | Spatiotemporal interpolation | $t \in [0.0, 0.5]$ | predefined in $t \in [0.0, 0.5]$ | grid at $t = 0.25$ | predefined in $t \in [0.0, 0.5]$ | grids at $t \in \{0.05, 0.15, 0.35, 0.45\}$ |
| | Temporal extrapolation | $t \in [0.0, 0.5]$ | grid at $t = 0.5$ | grid at $t = 0.6$ | grid at $t = 0.6$ | grids at $t \in [0.6, 1.0]$ |
| | Unseen coeff temporal extrapolation | - | - | - | grid at $t = 0.1$ | grids at $t \in [0.2, 0.5]$ |
| SWE (Section 4.3.1) | Spatiotemporal interpolation | $t \in [0.0, 0.5]$ | predefined in $t \in [0.0, 0.5]$ | grid at $t = 0.25$ | predefined in $t \in [0.0, 0.5]$ | grids at $t \in \{0.05, 0.15, 0.35, 0.45\}$ |
| | Temporal extrapolation | $t \in [0.0, 0.5]$ | grid at $t = 0.5$ | grid at $t = 0.6$ | grid at $t = 0.6$ | grids at $t \in [0.6, 1.0]$ |
| CNSE (Section 4.3.2) | Operator learning | $t \in \{0.0, 0.25, 0.5, 0.75, 1.0\}$ | grid at $t = 0.0$ | grids at $t \in \{0.25, 0.5, 0.75, 1.0\}$ | grid at $t = 0.0$ | grids at $t \in \{0.25, 0.5, 0.75, 1.0\}$ |
| Airfoil (Section H.2) | Operator learning | $t \in \{0.0, 1.2, 2.4, 3.6, 4.8\}$ | grid at $t = 0.0$ | grids at $t \in \{1.2, 2.4, 3.6, 4.8\}$ | grid at $t = 0.0$ | grids at $t \in \{1.2, 2.4, 3.6, 4.8\}$ |
| Darcy Flow (Section H.3) | Operator learning | | | $a \mapsto u$ | | |

## C    PINN Used in Prior Generation

In this study, we utilize the PINN introduced by Raissi et al. (2019a) to generate PINN priors. The loss function employed during the training of the PINN is as follows:

$$\mathcal{L} = \mathcal{L}_u + \mathcal{L}_f + \mathcal{L}_b,$$

where $\mathcal{L}_u, \mathcal{L}_f$ and $\mathcal{L}_b$ are defined as

$$\mathcal{L}_u = \frac{1}{N_u} \sum \left( \tilde{u}(x, 0) - u(x, 0) \right)^2, \quad \mathcal{L}_f = \frac{1}{N_f} \sum \left( \mathcal{N}(t, x, u, \alpha) \right)^2, \quad \mathcal{L}_b = \frac{1}{N_b} \sum \left( \tilde{u}(0, t) - \tilde{u}(2\pi, t) \right)^2,$$

for $N_u$ points at initial condition, $N_f$ collocation points, and $N_b$ boundary points.

The generation time for the PINN prior varies depending on the number of training epochs and the PINN loss threshold. Based on the configuration used in our experiments, it took approximately 373 seconds to generate the prior for a single coefficient combination of the 2D CDR and utilized up to 4,552 MB of memory.

## D    Detailed Description of the Baselines

We provided descriptions of each baseline discussed in Section 4.1.

- DeepONet is a neural operator architecture designed to learn operators mapping input functions to output functions. It combines branch and trunk networks to predict values in a function space.
- FNO learns solution operators for partial differential equations (PDEs) using the Fourier transform. By mapping inputs to a frequency domain, FNO captures complex patterns and long-range dependencies and models complex systems.

- F-FNO extends the Fourier Neural Operator by factorizing its layers to reduce computational costs. This factorization enables efficient learning of solution operators for complex systems.

- A-FNO is a variant of the Fourier Neural Operator that dynamically adjusts the resolution of the frequency domain during training. This adaptation aims to capture relevant features across scales, enabling more flexible modeling of complex systems.

- Poseidon is a multiscale operator transformer architecture designed to model solution operators for PDEs with continuous-in-time capability. It introduces time-conditioned layer normalization and processes multi-resolution input features to capture complex spatiotemporal dynamics.

- DPOT is a scalable neural operator model designed for PDEs, featuring a Fourier attention mechanism that enables efficient learning of complex spatiotemporal dynamics. Its architecture incorporates an auto-regressive denoising pre-training strategy, facilitating stable and flexible modeling across diverse PDE datasets with varying resolutions and geometries.

In the spatiotemporal interpolation task, models other than DeepONet are excluded from comparison because they cannot process mesh-structured inputs. DeepONet requires input data from fixed positions for each dataset; therefore, for a fair comparison, our model is also provided with context data from the same positions during evaluation. For Poseidon and DPOT, which are scientific foundation models providing pre-trained weights, we utilized their released weights and performed fine-tuning. For a fair comparison, we trained them and our model using only half the number of epochs compared to the other models. Furthermore, computational costs, memory usage, experiment settings, and hyperparameters are comprehensively outlined in Appendix I.

Table 7: Major comparisons between baselines and our model on the shape of input and target data (context and query in our model). Neural operator baselines (excluding DeepONet) can only predict targets at predetermined locations and are limited to processing grid-structured input data. DeepONet, on the other hand, can handle inputs of various shapes but requires fixed-coordinate input values, though it can produce target solutions at desired coordinates. In contrast, our model imposes no restrictions on the shape, coordinates, or number of inputs and can generate target solutions at arbitrary coordinates.

| Data | FNO & F-FNO & A-FNO & Poseidon & DPOT | DeepONet | Ours |
|---|---|---|---|
| Input data | Grid of predefined coordinate | Mesh of predefined coordinate | **Mesh of arbitrary coordinate** |
| Target data | Grid of predefined coordinate | **Mesh of arbitrary coordinate** | **Mesh of arbitrary coordinate** |

## E  MODEL ARCHITECTURE

Our model is fundamentally based on the Prior-Fitted Network (PFN) architecture (Müller et al., 2022), and consists of three main components: an encoder, a Transformer block, and a decoder. In this section, we describe the encoder. For clarity, we denote the training contexts and queries as $D$ and $T$, respectively, consistent with the main text. Each input in $D$ and $T$ is composed of spatial coordinates $x$, temporal coordinates $t$, and solution values $y$; for queries $T$, the solution values are masked. The encoder enriches these inputs through a Fourier feature embedding (FFE) followed by a multilayer perceptron (MLP).

**Encoder**  The role of FFE is to augment the raw inputs with high-frequency components, enabling the model to better capture complex solution patterns. This advantage of using FFE in conjunction with MLPs has been verified in (Tancik et al., 2020; Ma et al., 2025). For grid-based inputs, we can employ a discrete Fourier transform via fast Fourier transform (FFT). For coordinate-based inputs, we use sinusoidal feature mappings with predetermined frequencies. Specifically, given an input $v \in \mathbb{R}$, we define the embedding as

$$\text{FFE}(v; n) = [sin(v \cdot \omega_1), cos(v \cdot \omega_1), \cdots, sin(v \cdot \omega_n), cos(v \cdot \omega_n)],$$

where the frequency coefficients are set as $\omega_i = 2\pi/i$. After applying FFE, the encoder constructs three types of feature vectors: $l_{CD} := [x_D, t_D, \text{FFE}(x_D), \text{FFE}(t_D)]$ for context domain information,

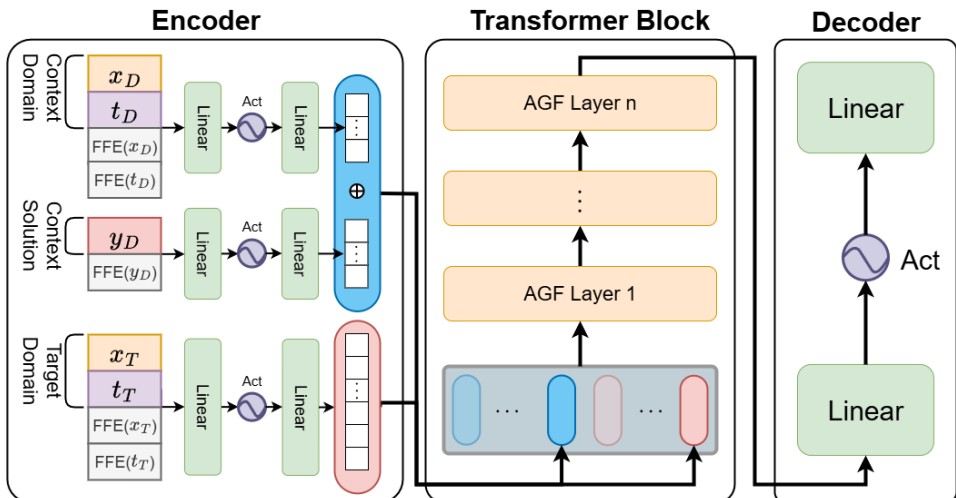

Figure 3: A diagram of PDE-PFN's architecture. The diagram is presented as an extension of Figure 2. The model consists of three main components: an encoder, a Transformer block, and a decoder. FFE denotes the Fourier feature embedding, and Act denotes the activation function.

$l_{CS} := [y_D, \text{FFE}(y_D)]$ for context solutions, and $l_{QD} := [x_T, t_T, \text{FFE}(x_T), \text{FFE}(t_T)]$ for query domain information, by concatenating the raw inputs and FFE results. Each of these vectors is passed through a separate MLP to learn PDE-specific embeddings. Each MLP consists of two linear layers and a single activation function between linear layers. For the activation, we employ a rational activation function (Molina et al., 2019), which has an adaptive nature and has been shown to provide greater flexibility compared to conventional nonlinearities. Finally, $l_{CD}$ and $l_{CS}$ are concatenated after passing each MLP to form the context representation used by the Transformer block.

**Transformer block** The encoder output is then processed through a Transformer block, which consists of a sequence of Transformer layers. Instead of the *vanilla* Transformer layer, we adopt the attentive graph filter (AGF) layer Wi et al. (2025), which provides greater flexibility while reducing computational complexity. The AGF layer enhances standard linear attention by learning attention in the singular value domain. Specifically, it decomposes the attention operation into a set of spectral components and adaptively learns their importance, enabling the model to capture the dependencies more effectively. Specifically, the attention matrix $A \in \mathbb{R}^{n \times m}$ can be decomposed using singular value decomposition (SVD) as $A = U\Sigma V^T$, where $U$ and $V$ are orthogonal matrices, and $\Sigma = \text{diag}(\sigma_1, \sigma_2, \ldots, \sigma_r)$ contains the singular values. Therefore, as discussed in Section 3, when using the AGF layer, we include a regularization term to ensure the orthogonality of the matrices $U$ and $V$. The regularization term is defined as follows:

$$\mathcal{L}_{\text{AGF}} = \frac{1}{n^2}\left(\left\|U^\top U - \mathbf{I}\right\|\right) + \frac{1}{m^2}\left(\left\|VV^\top - \mathbf{I}\right\|\right),$$

where $\mathbf{I}$ is identity matrix. AGF learns to adaptively reweight these spectral components, effectively filtering the most informative modes while suppressing noise or redundant information. By learning attention in the singular value domain, AGF provides a more powerful representation of global dependencies compared to conventional linear attention, while avoiding the quadratic complexity of the *vanilla* Transformers.

To further enrich the model capacity, we separate the parameters used for self-attention and cross-attention within each layer. As illustrated in Figure 2, self-attention is applied among the contexts $D$, while cross-attention allows queries $T$ to attend to contexts $D$. In the figure, self-attention and cross-attention are represented by red and blue arrows, respectively. The numbers of blue rods $D$ and red rods $T$ are described in Table 20 in Appendix I.

**Decoder** The output of the Transformer block is finally passed through a decoder, implemented as a simple MLP. Similar to the encoder, this MLP consists of two linear layers and a single

activation function between linear layers. The decoder maps the latent representations produced by the Transformer into the predicted solution values at the queried coordinates.

## F  TRAINING

Unlike other task-specific models, our model is trained in an integrated manner, enabling it to solve multiple tasks simultaneously in that dataset. For cases where both spatiotemporal interpolation and temporal extrapolation tasks must be evaluated, such as in the family of 2D CDR equations and SWE datasets, we perform training with temporal extrapolation as the primary objective. In this setup, other baselines treat grid data points at $t_n$ and $t_{n+1}$ as fixed input–output pairs. In contrast, our model constructs pairs by randomly mixing the data points within each $(t_n, t_{n+1})$ pair independently, rather than preserving their exact correspondence. This strategy enables the model to learn in a way that supports both spatiotemporal interpolation and temporal extrapolation within a unified framework. For the CNSE, Airfoil, and Darcy Flow datasets, where we conducted experiments only on operator learning, random mixing was not applied.

**Training algorithm**  We train our model as follows:

---

**Algorithm 1** Training our model

---

1: **Input:** contexts $D$ and queries $T$ from prior $p(\mathcal{D})$ in dataset
2: **Output:** Our model $\tilde{u}_\theta$ which can approximate the PPD
3: Initialize the model $\tilde{u}_\theta$
4: **for** $i = 1$ to $n$ **do**
5:     Sample $\boldsymbol{\alpha} \in \Omega$ and $D, T \subseteq \tilde{u}(\boldsymbol{\alpha}) \sim p(\mathcal{D})$ ($D := \{(x_D^{(i)}, t_D^{(i)})\}_{i=1}^{N_D}$, $T := \{(x_T^{(j)}, t_T^{(j)})\}_{j=1}^{N_T}$)
6:     **if** dataset $\in$ {Family of 2D CDR, SWE} **then**
7:         Regenerate $D'$ and $T'$ by shuffling data points in $D \cup T$ independently within each $(t_n, t_{n+1})$
            pair, while preserving $|D'| = N_D$ and $|T'| = N_T$
8:     **end if**
9:     Compute MSE loss $\mathcal{L}_{\boldsymbol{\alpha}} = \frac{1}{N_T} \sum_{j=1}^{N_T} \left\{ \tilde{u}_\theta(x_T^{(j)}, t_T^{(j)} | D_n') - \tilde{u}(x_T^{(j)}, t_T^{(j)}) \right\}^2$
10:     Compute AGF regularization term $\mathcal{L}_{AGF}$ and objective function $\mathcal{L} = \mathcal{L}_{\boldsymbol{\alpha}} + \mathcal{L}_{AGF}$
11:     Update parameters $\theta$ with an AdamW optimizer
12: **end for**

---

## G  EVALUATION

We employ $L_1$ absolute, $L_2$ relative, and $L_\infty$ relative errors between the model's prediction for test queries and the ground truth. The errors are then averaged over the target parameter space or test dataset. Given the true solution $y_{\alpha,k}$ and the corresponding prediction $\hat{y}_{\alpha,k}$ at the $k$-th time point out of a total of $K$ evaluation time points, each metric is computed as follows:

$$L_p \text{ abs error} = \frac{1}{|\Omega| \cdot K} \sum_{\alpha \in \Omega} \sum_{k=1}^{K} ||y_{\alpha,k} - \hat{y}_{\alpha,k}||_p \, , \; L_p \text{ rel error} = \frac{1}{|\Omega| \cdot K} \sum_{\alpha \in \Omega} \sum_{k=1}^{K} \frac{||y_{\alpha,k} - \hat{y}_{\alpha,k}||_p}{||y_{\alpha,k}||_p}.$$

## H  ADDITIONAL EXPERIMENTS

### H.1  TEST TIME EVALUATION GIVEN PINN PRIOR IN THE FAMILY OF 2D CDR EQUATIONS

As an additional experiment on the family of 2D CDR equations, we modify the test procedure from Section 4.2.1. While keeping all other experimental settings the same, we compare the evaluation results when noisy PINN priors are provided as input during testing. This experiment is designed to assess the robustness of the models in producing accurate solutions despite noisy inputs.

As shown in the experimental results presented in Table 8, Ours achieves the best performance, demonstrating strong robustness. For Ours (PINN), however, the performance in terms of two relative errors on the spatiotemporal interpolation task is inferior to that of DeepONet. This can be attributed

Table 8: The evaluation results for the spatiotemporal interpolation and temporal extrapolation task applied to the family of 2D CDR equations given a noisy PINN prior at evaluation. They are measured at the seen coefficients ($\beta_x, \beta_y, \nu_x, \nu_y, \rho_1, \rho_2, \rho_3 \in \{0.0, 0.5, 1.0\}$). The best performance is marked in **bold** and the second-best performance is marked with underline.

| Task | Metric | Ours | Ours (PINN) | DeepONet | FNO | F-FNO | A-FNO | LNO | Poseidon | DPOT |
|---|---|---|---|---|---|---|---|---|---|---|
| Spatiotemporal interpolation | $L_1$ Abs | **0.01953** | 0.02445 | 0.02540 | - | - | - | - | - | |
| | $L_2$ Rel | **0.02755** | 0.03467 | 0.03413 | - | - | - | - | - | - |
| | $L_\infty$ Rel | 0.10158 | 0.10716 | **0.09286** | - | - | - | - | - | - |
| Temporal extrapolation | $L_1$ Abs | **0.01655** | 0.02099 | 0.06888 | 0.03458 | 0.02816 | 0.05794 | 0.03048 | 0.14018 | 0.05051 |
| | $L_2$ Rel | **0.02486** | 0.02893 | 0.08591 | 0.04268 | 0.03582 | 0.07903 | 0.03907 | 0.15380 | 0.08624 |
| | $L_\infty$ Rel | 0.08830 | **0.09333** | 0.23386 | 0.10203 | 0.10677 | 0.28805 | 0.11730 | 0.28481 | 0.08624 |

to the fact that Ours (PINN) is pre-trained to predict solutions from PINN priors as inputs, which may cause the model to generate outputs closer to the PINN prior rather than the analytic solution when the prior is provided at test time. Nevertheless, since Ours (PINN) still achieves better performance in terms of $L_1$ absolute error, we can conclude that it also exhibits a certain degree of robustness.

## H.2 OPERATOR LEARNING IN AIRFOIL

To evaluate input flexibility in operator learning, we conduct experiments on the Airfoil dataset defined on an irregular mesh structure. This dataset consists of 2,000 samples, of which 700 are used for training, 100 for validation, and 200 for testing. The target operator is formulated similarly to that in Section 4.3.2: given the initial condition, the model is required to predict the solutions at four time steps, $t = 1.2, 2.4, 3.6,$ and $4.8$ (see Appendix B.4 for a formal description). Most of the baselines used in Section 4 rely on data defined on a regular grid, so that cannot be applied here, since the Airfoil dataset is defined on an irregular mesh. Consequently, among the existing baselines, we only use DeepONet and additionally include Oformer (Li et al., 2023) as a new baseline.

Table 9: The evaluation results for the operator learning task applied to the Airfoil dataset. The best performance is marked in **bold** and the second-best performance is marked with underline.

| Metric | Ours | Ours (PINN) | DeepONet | Oformer |
|---|---|---|---|---|
| $L_1$ Abs | 0.11047 | **0.10755** | 0.23160 | 0.12039 |
| $L_2$ Rel | 0.18831 | **0.18357** | 0.44101 | 0.19744 |
| $L_\infty$ Rel | 0.33584 | 0.32887 | 0.73076 | **0.32318** |

The difference from the experimental results reported in the original Oformer paper arises because their setting involved using the first four time steps as input to predict subsequent time steps, whereas our experiment is formulated based on the initial condition only. The experimental results in Table 9 show that both versions of our model outperform the baselines on two metrics. These results demonstrate that our model's ICL capability effectively extends to irregular mesh data and can generalize well to new PDE problems. Detailed information on the best hyperparameter settings can be found in Appendix I.3.

## H.3 OPERATOR LEARNING IN DARCY FLOW

To further evaluate the PDE generalization ability of our model in operator learning, we conducted additional experiments on the Darcy Flow dataset. This dataset consists of 2,000 samples, with 1,400 used for training, 200 for validation, and 400 for testing. The operator learning task is defined as predicting the steady-state solution $u(x)$ from a given viscosity field $a(x)$ (see Appendix B.5 for a formal description). Unlike the CNSE dataset, where both context and query represent the same feature, the Darcy Flow dataset involves mapping between different quantities. This makes it a distinct form of operator learning task.

Table 10: The evaluation results for the operator learning task applied to Darcy Flow. The best performance is marked in **bold** and the second-best performance is marked with underline.

| Metric | Ours | Ours (PINN) | DeepONet | FNO | F-FNO | A-FNO | Poseidon | DPOT |
|---|---|---|---|---|---|---|---|---|
| $L_1$ Abs | 0.00796 | 0.00742 | 0.00996 | 0.00792 | 0.00806 | 0.00906 | 0.00998 | **0.00726** |
| $L_2$ Rel | 0.28771 | **0.28091** | 0.40759 | 0.29292 | 0.34149 | 0.40357 | 0.41001 | 0.51505 |
| $L_\infty$ Rel | **0.39694** | 0.40124 | 0.59057 | 0.46398 | 0.59381 | 0.80674 | 0.52933 | 0.51505 |

The experimental results in Table 10 show that both versions of our model outperform the baselines on two metrics. These findings further confirm that our model maintains strong PDE generalization ability even when the operator learning task requires transferring between different physical quantities. Detailed information on the best hyperparameter settings can be found in Appendix I.3.

### H.4 SENSITIVITY ANALYSIS TO THE NUMBER OF TEST CONTEXT POINTS IN THE FAMILY OF 2D CDR EQUATION

In this section, we provide empirical validation of Theorem 2.1, which establishes the theoretical consistency of the neural network's posterior predictive distribution (PPD) as the size of the data $D_n$ increases. Specifically, we evaluate the sensitivity of the neural network's PPD to the number of the context set $\overline{D}$ provided during the test process.

To this end, we conduct an evaluation of spatiotemporal interpolation on the family of 2D CDR equations experiment by varying the number of the context set $\overline{D}$ while keeping other factors unchanged, including the best hyperparameter settings. The results, depicted in Figure 4, clearly demonstrate that as the size of $\overline{D}$ increases, the error consistently decreases across all three evaluation metrics. This behavior aligns perfectly with the theoretical prediction in Theorem 2.1, where the posterior approximation is shown to converge toward the true distribution as the data size grows.

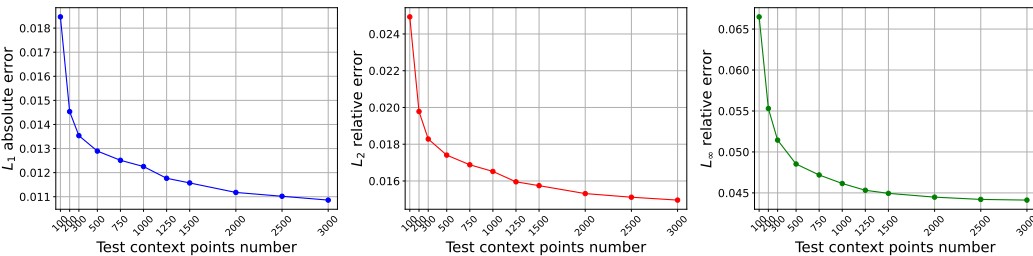

Figure 4: Sensitivity to the number of the context data $\overline{D}$ during the test.

These experimental findings not only validate the theoretical insights of Theorem 2.1 but also emphasize the robustness and accuracy of the neural network's PPD approximation under the given prior. The decreasing error trend highlights how the model effectively integrates increasing amounts of data to produce predictions that are more consistent with the true underlying posterior distribution. This synergy between theory and empirical observation strongly supports the reliability and effectiveness of the proposed approach.

### H.5 UNSEEN DOMAIN TEMPORAL EXTRAPOLATION FOR UNSEEN COEFFICIENT 2D CDR EQUATIONS

In the experiments on unseen coefficients presented in the main text, the spatiotemporal domain was restricted to the temporal range used during training. This design choice ensured that only the parameter domain included unseen values, preventing both parameter and temporal extrapolation from occurring simultaneously. In this section, we extend the evaluation to a more challenging setting in which both the parameter domain and the temporal domain involve unseen ranges. The unseen coefficient ranges follow those used in Section 4.2.2, and for the temporal domain we adopt the evaluation range from the temporal extrapolation task in Section 4.2.1.

Table 11: The evaluation results for the temporal extrapolation tasks applied to the family of 2D CDR equations. They are measured at the unseen coefficients and unseen temporal domain. The best performance is marked in **bold** and the second-best performance is marked with underline.

| Coefficients | Task | Metric | Ours | Ours (PINN) | DeepONet | FNO | F-FNO | A-FNO | LNO | Poseidon | DPOT |
|---|---|---|---|---|---|---|---|---|---|---|---|
| Interpolation | Temporal extrapolation | $L_1$ Abs | **0.00736** | 0.00939 | 0.05139 | 0.10863 | 0.01405 | 0.04695 | 0.01839 | 0.13852 | 0.04610 |
| | | $L_2$ Rel | **0.00954** | 0.01178 | 0.06179 | 0.14380 | 0.01636 | 0.06227 | 0.02049 | 0.14993 | 0.04676 |
| | | $L_\infty$ Rel | **0.03193** | 0.03693 | 0.16008 | 0.34858 | 0.03847 | 0.21750 | 0.04512 | 0.26293 | 0.04676 |
| Extrapolation | Temporal extrapolation | $L_1$ Abs | **0.01027** | 0.02063 | 0.05514 | 0.03887 | 0.02312 | 0.05013 | 0.02560 | 0.14661 | 0.05186 |
| | | $L_2$ Rel | **0.01068** | 0.02169 | 0.06450 | 0.05114 | 0.02470 | 0.06351 | 0.02677 | 0.15650 | 0.05189 |
| | | $L_\infty$ Rel | **0.01661** | 0.03474 | 0.16555 | 0.13713 | 0.04591 | 0.20955 | 0.04911 | 0.27502 | 0.05189 |

The experimental results in Table 11 show that our model continues to achieve the best performance. In contrast, the baselines experience greater difficulty when both types of extrapolation are required simultaneously, performing noticeably worse than our model in this setting.

## H.6 PDE GENERALIZATION FOR 3D DIFFUSION-REACTION EQUATIONS

To further explore the potential of our model for 3D PDEs, we conduct additional experiments on 3D diffusion–reaction equations. We construct a 3D diffusion–reaction equation dataset by extending the same procedure used for generating the 2D CDR equations. This formulation includes three diffusion coefficients $(\nu_x, \nu_y, \nu_z)$ and one reaction coefficient $\rho$, each sampled from $0.0, 0.5, 1.0, 1.5$, resulting in 256 distinct equations. We perform experiments on the same two tasks as in the 2D CDR equations—spatiotemporal interpolation and temporal extrapolation—using DeepONet, FNO, and DPOT as baselines, since only they can handle 3D datasets among baselines.

Table 12: The evaluation results for the spatiotemporal interpolation and temporal extrapolation tasks applied to the 3D diffusion reaction equations. They are measured at the seen coefficients $(\nu_x, \nu_y, \nu_z, \rho_1 \in \{0.0, 0.5, 1.0, 1.5\})$. The best performance is marked in **bold** and the second-best performance is marked with underline.

| Task | Metric | Ours | Ours (PINN) | DeepONet | FNO | DPOT |
|---|---|---|---|---|---|---|
| Spatiotemporal interpolation | $L_1$ Abs | **0.02685** | 0.02843 | 0.02816 | | |
| | $L_2$ Rel | **0.03711** | 0.04144 | 0.03981 | - | - |
| | $L_\infty$ Rel | **0.10938** | 0.12823 | 0.12724 | | |
| Temporal extrapolation | $L_1$ Abs | **0.01031** | 0.01039 | 0.02421 | 0.01521 | 0.01485 |
| | $L_2$ Rel | **0.01376** | 0.01391 | 0.03033 | 0.01824 | 0.01638 |
| | $L_\infty$ Rel | 0.04180 | 0.04545 | 0.11530 | 0.05545 | **0.01638** |

The experimental results in Table 12 show that our model outperforms the baselines in spatiotemporal interpolation in all metrics and in temporal extrapolation on two metrics, demonstrating its potential to generalize effectively to 3D datasets.

## H.7 ABLATION STUDY

In this section, we conduct an ablation study to analyze the necessity of the key components in our model. Using the 2D CDR equation experiments, we replace AGF with a *vanilla* Transformer layer and substitute the rational activation with GELU to evaluate the impact of each component on performance.

Table 13: The evaluation results for the spatiotemporal interpolation and temporal extrapolation tasks applied to the family of 2D CDR equations. "Ours w.o. agf," "Ours w.o. act," and "Ours w.o. all" denote replacing AGF with a vanilla Transformer block, replacing the rational activation with GELU, and replacing both, respectively. They are measured at the seen coefficients $(\beta_x, \beta_y, \nu_x, \nu_y, \rho_1, \rho_2, \rho_3 \in \{0.0, 0.5, 1.0\})$. The best performance is marked in **bold** and the second-best performance is marked with underline.

| Task | Metric | Ours | Ours w.o. agf | Ours w.o. act | Ours w.o. all |
|---|---|---|---|---|---|
| Spatiotemporal interpolation | $L_1$ Abs | 0.01218 | 0.01242 | **0.01040** | 0.01780 |
| | $L_2$ Rel | 0.01643 | 0.01625 | **0.01495** | 0.02281 |
| | $L_\infty$ Rel | 0.04600 | **0.04297** | 0.04399 | 0.05736 |
| Temporal extrapolation | $L_1$ Abs | **0.01474** | 0.01776 | 0.01725 | 0.02454 |
| | $L_2$ Rel | **0.02261** | 0.02779 | 0.02665 | 0.03437 |
| | $L_\infty$ Rel | **0.08444** | 0.09554 | 0.09376 | 0.10625 |

The experimental results in Table 13 show that although the ablated variants may occasionally perform slightly better on the interpolation task, our full model consistently achieves superior performance on the more challenging temporal extrapolation task. This supports our claim that the proposed architectural components enhance the model's ability to generalize across diverse tasks in a stable and efficient manner.

We additionally examine whether the robustness observed in Section 4.2 is preserved by conducting the same PINN-prior evaluation on the "Ours with sinusoidal activation'' and "Ours w.o. AGF'' variants.

Table 14: The evaluation results for the spatiotemporal interpolation and temporal extrapolation tasks applied to the family of 2D CDR equations. "Ours w.o. agf," "Ours w.o. act," and "Ours w.o. all" denote replacing AGF with a vanilla Transformer block, replacing the rational activation with GELU, and replacing both, respectively. They are measured at the seen coefficients $(\beta_x, \beta_y, \nu_x, \nu_y, \rho_1, \rho_2, \rho_3 \in \{0.0, 0.5, 1.0\})$. The best performance is marked in **bold** and the second-best performance is marked with underline.

| Task | Metric | Ours | Ours w.o. agf | Ours with sinusoidal |
|------|--------|------|---------------|---------------------|
| Spatiotemporal interpolation | $L_1$ Abs | **0.01507** | 0.01667 | 0.01482 |
| | $L_2$ Rel | **0.02057** | 0.02171 | 0.02096 |
| | $L_\infty$ Rel | 0.06584 | **0.05536** | 0.06946 |
| Temporal extrapolation | $L_1$ Abs | **0.01940** | 0.02105 | 0.02671 |
| | $L_2$ Rel | **0.02705** | 0.03030 | 0.04159 |
| | $L_\infty$ Rel | **0.08802** | 0.09518 | 0.14922 |

Table 14 shows that our model achieves the best results on both tasks, confirming strong robustness, whereas both ablations exhibit degraded performance. These results further indicate that the AGF layer and rational activation function contribute significantly to the model's robustness.

## H.8 TEST TIME EVALUATION GIVEN PRIOR WITH BIASED NOISE IN THE FAMILY OF 2D CDR EQUATIONS

Table 15: The evaluation results for the temporal extrapolation tasks applied to the family of 2D CDR equations. They are measured at the seen coefficients $(\beta_x, \beta_y, \nu_x, \nu_y, \rho_1, \rho_2, \rho_3 \in \{0.0, 0.5, 1.0\})$.

| Bias | No noise | 0.1% | 0.5% | 1.0% | 1.5% | 2.0% | 2.5% | 3.0% | 3.5% | 4.0% |
|------|----------|------|------|------|------|------|------|------|------|------|
| $L_1$ Abs | 0.01474 | 0.01483 | 0.01472 | 0.01481 | 0.01514 | 0.01572 | 0.01664 | 0.01800 | 0.01985 | 0.02212 |
| $L_2$ Rel | 0.02261 | 0.02271 | 0.02246 | 0.02244 | 0.02271 | 0.02330 | 0.02255 | 0.02573 | 0.02772 | 0.03009 |
| $L_\infty$ Rel | 0.08444 | 0.08556 | 0.08459 | 0.08426 | 0.08490 | 0.08646 | 0.08901 | 0.09265 | 0.09733 | 0.10276 |

Since Theorem 2.1 assumes unbiased noise, we further investigate how robust our model remains when the inference stage includes biased Gaussian noise added to the analytic solution. We set the standard deviation of Gaussian noise to $\sigma = 0.01$ and vary only the magnitude of the bias. The evaluation is performed on the temporal extrapolation task of the 2D CDR equations, using the $L_2$ relative error for comparison.

The experimental results in Table 15 show that when the bias is below 3%, the performance degradation remains relatively minor compared to the bias level. However, once the bias reaches approximately 3% or higher, the error begins to increase more noticeably. This indicates that PDE-PFN maintains robustness against moderate systematic bias, but large bias eventually leads to a degradation in accuracy as expected.

## H.9 COMPUTATIONAL COST ANALYSIS RELATED TO THE NUMBER OF CONTEXT AND QUERY POINTS

In this section, we analyze how the number of context and query points affects the training and inference time as well as memory cost. For both training and inference, we report the average per-iteration runtime measured over 10 iterations and record the peak GPU memory usage. When evaluating the effect of context size, we fix the number of query points to 4,096. Likewise, when evaluating the effect of query size, we fix the number of context points to 4,096.

Table 16: Training and inference performance with different **context lengths**.

| Context length | 1,024 | 2,048 | 4,096 | 8,192 | 12,288 | 16,384 |
|----------------|-------|-------|-------|-------|--------|--------|
| Training time (sec) | 0.104 | 0.102 | 0.101 | 0.107 | 0.136 | 0.165 |
| Training memory (MB) | 280.0 | 329.8 | 420.7 | 617.2 | 800.4 | 987.4 |
| Inference time (sec) | 0.030 | 0.041 | 0.065 | 0.115 | 0.165 | 0.213 |
| Inference memory (MB) | 237.8 | 312.0 | 459.7 | 755.7 | 1054.7 | 1349.7 |

As shown in Tables 16 and 17, all four metrics increase linearly as the number of context or query points increases. This confirms that, due to our use of AGF layers, the model exhibits linear rather than quadratic complexity, unlike a *vanilla* Transformer.

Table 17: Training and inference performance with different **query lengths**.

| Query length | 1,024 | 2,048 | 4,096 | 8,192 | 12,288 | 16,384 |
|---|---|---|---|---|---|---|
| Training time (sec) | 0.101 | 0.100 | 0.101 | 0.123 | 0.148 | 0.176 |
| Training memory (MB) | 872.7 | 1010.1 | 1265.4 | 1788.6 | 2303.0 | 2831.2 |
| Inference time (sec) | 0.056 | 0.059 | 0.065 | 0.078 | 0.090 | 0.104 |
| Inference memory (MB) | 358.3 | 391.7 | 458.7 | 594.7 | 733.7 | 867.7 |

## H.10 2D CDR EQUATIONS WITH DIFFERENT INITIAL CONDITION

In this experiment, we examine whether our model continues to perform well on the 2D CDR equations from the main paper when the dataset is generated using a different initial condition. Specifically, we replace the original initial condition, $1 + \sin(2x)\sin(2y)$, with $1 + \sin(3x)\sin(3y)$ to construct a new 2D CDR equation dataset. We then conduct training and evaluation following the same procedure used in the main experiments.

Table 18: The evaluation results for the spatiotemporal interpolation and temporal extrapolation tasks applied to the new family of 2D CDR equations. They are measured in the same way of Section 4.2.1. The best performance is marked in **bold** and the second-best performance is marked with underline.

| Task | Metric | Ours | DeepONet | FNO | F-FNO | A-FNO | LNO | Poseidon | DPOT |
|---|---|---|---|---|---|---|---|---|---|
| Spatiotemporal interpolation | $L_1$ Abs | **0.01339** | 0.07394 | - | - | - | - | - | - |
| | $L_2$ Rel | **0.01885** | 0.09376 | | | | | | |
| | $L_\infty$ Rel | **0.06606** | 0.26320 | | | | | | |
| Temporal extrapolation | $L_1$ Abs | **0.02017** | 0.03988 | 0.03184 | 0.03371 | 0.08342 | 0.04696 | 0.09397 | 0.12628 |
| | $L_2$ Rel | **0.02929** | 0.05282 | 0.04160 | 0.04192 | 0.09967 | 0.06282 | 0.10352 | 0.13633 |
| | $L_\infty$ Rel | **0.10793** | 0.17487 | 0.12671 | 0.10980 | 0.27716 | 0.18313 | 0.23855 | 0.13633 |

The experimental results in Table 18 show that our model continues to achieve the best performance even when evaluated on datasets generated with a new initial condition. This confirms that our model does not merely perform well on a single fixed initial condition, but instead generalizes robustly across different initial conditions. Furthermore, the experiments on the SWE and CNSE datasets in the main paper also support this conclusion, as both datasets consist of samples generated from varying initial conditions under the same underlying PDE. Thus, our model is not limited to datasets constructed with a single, unified initial condition.

## I EXPERIMENTAL DETAILS

### I.1 ENVIRONMENTS

The experiments on the family of 2D CDR equations, SWE, CNSE, and Darcy Flow were conducted using an NVIDIA RTX A6000, while the CNSE experiments were conducted on an NVIDIA RTX A5000. Details of the Python version and the packages used in the experiments can be found in the accompanying code's environment file.

### I.2 COST COMPARISON ON THE MAIN EXPERIMENTS

In addition to the comparison between the baselines in Table 7, the additional comparison between the baselines is shown below in Table 19. In this tables, we compare the number of parameters, inference time, and the GPU memory usage for all models in their best settings. The inference time and GPU memory usage are measured in the test process on the family of 2D CDR equations temporal extrapolation task. In the experiment, both versions of our model are run with the same hyperparameters; therefore, we report the results under the single label Ours.

### I.3 HYPERPARAMETERS

In this section, we describe the number of data points for each equation and the hyperparameters used in the experiments for each model. In Table 20, the data points used in each experiment are shown separately for training, validation, and testing. For the training data, the context and query sets are constructed differently depending on the task. In the family of 2D CDR equations experiments

Table 19: Additional comparisons between baselines and our model in the family of 2D CDR equations temporal extrapolation test process.

| Model | Number of parameters | inference time per sample(s) | GPU memory usage(MB) |
|---|---|---|---|
| Ours | 4,488,873 | 0.24 | 26.64 |
| DeepONet | 723,969 | 0.01 | 11.06 |
| FNO | 143,268,097 | 0.10 | 1098.86 |
| F-FNO | 23,631,105 | 0.02 | 98.47 |
| A-FNO | 7,155,472 | 0.08 | 35.62 |
| Poseidon | 157,625,930 | 0.27 | 2408.40 |
| DPOT | 475,926,558 | 0.20 | 1825.72 |

with unseen coefficients, we only performed testing without training or validation; therefore, only the testing data points are reported.

Table 20: Number of data points used for training, validation, and testing in all experiments.

| PDE | Task | Train | Valid | | Test | |
|---|---|---|---|---|---|---|
| | | | Context | Query | Context | Query |
| Family of 2D CDR equations | Spatiotemporal interpolation | 6,144 | 1,024 | 1,024 | 1,024 | 4,096 |
| | Temporal extrapolation | 6,144 | 1,024 | 1,024 | 1,024 | 4,096 |
| | Unseen coeff temporal extrapolation | - | - | - | 1,024 | 4,096 |
| SWE | Spatiotemporal interpolation | 98,304 | 16,384 | 16,384 | 16,384 | 65,536 |
| | Temporal extrapolation | 98,304 | 16,384 | 16,384 | 16,384 | 65,536 |
| CNSE | Time trajectory predicting operator learning | 20,480 | 4,096 | 16,384 | 4,096 | 16,384 |
| Airfoil | Time trajectory predicting operator learning | 26,165 | 5,233 | 20,932 | 5,233 | 20,932 |
| Darcy Flow | $a \mapsto u$ operator learning | $a : 16,384$ $u : 16,384$ | $a : 16,384$ $u : 16,384$ | $a : 16,384$ $u : 16,384$ | $a : 16,384$ $u : 16,384$ | $a : 16,384$ $u : 16,384$ |

In the family of 2D CDR equation experiments, the evaluations on unseen coefficients are conducted without additional training. Therefore, we report only the hyperparameters used for the spatiotemporal interpolation and temporal extrapolation tasks trained on seen coefficients. During fine-tuning after pre-training, the first linear layer of the encoder and the final linear layer of the decoder are re-initialized to deal with cases where the number of variables to be predicted differs. For the family of 2D CDR equations, SWE, CNSE, and Airfoil experiments, we employ sinusoidal FFE, whereas FFT-based embeddings are used in the Darcy Flow experiments. For fairness, all best hyperparameters were selected based on those that achieved the highest performance on the validation. The hyperparameters for our model and the baselines in each experiment can be found in Table from 21 to 23. In each experiment, both versions of our model are run with the same hyperparameters; therefore, we report the results under the single label Ours.

Table 21: Best hyperparameter for each model used in the family of 2D CDR equations and SWE experiments.

| Model | Hyperparameter Name | Family of 2D CDR equations | | SWE | |
|---|---|---|---|---|---|
| | | Spatiotemporal interpolation | Temporal extrapolation | Spatiotemporal interpolation | Temporal extrapolation |
| Ours | Attention layers number | | | 16 | |
| | Attention hidden dimension | | | 64 | |
| | Attention head number | | | 5 | |
| | FFE dimension | | | 0 | |
| | AGF depth | | | 15 | |
| | En/Decoder hidden dimension | | | 512 | |
| DeepONet | Branch net depth | 6 | 5 | 5 | 6 |
| | Trunk net depth | 5 | 5 | 6 | 6 |
| | Hidden dimension | 256 | 256 | 256 | 256 |
| FNO | Layers number | - | 4 | - | 3 |
| | Hidden dimension | - | 256 | - | 128 |
| F-FNO | Layers number | - | 5 | - | 3 |
| | Hidden dimension | - | 256 | - | 128 |
| A-FNO | Layers number | - | 12 | - | 3 |
| | Hidden dimension | - | 256 | - | 256 |
| Poseidon | Pre-trained weight | - | L | - | T |
| DPOT | Pre-trained weight | - | L | - | M |
| PINN prior | Training loss threshold | $1 \times 10^{-4}$ | | - | |
| | Maximum training epoch | 200 | | - | |

Table 22: Best hyperparameter for each model used in the CNSE and Darcy Flow experiments.

| Model | Hyperparameter Name | CNSE | Darcy Flow |
|---|---|---|---|
| Ours | Attention layers number | | 16 |
| | Attention hidden dimension | | 64 |
| | Attention head number | | 5 |
| | FFE dimension | 0 | 3 |
| | AGF depth | | 15 |
| | En/Decoder hidden dimension | | 512 |
| DeepONet | Branch net depth | 6 | 6 |
| | Trunk net depth | 5 | 5 |
| | Hidden dimension | 256 | 256 |
| FNO | Layers number | 3 | 6 |
| | Hidden dimension | 256 | 128 |
| F-FNO | Layers number | 4 | 3 |
| | Hidden dimension | 256 | 256 |
| A-FNO | Layers number | 8 | 16 |
| | Hidden dimension | 256 | 128 |
| Poseidon | pre-trained weight | T | T |
| DPOT | pre-trained weight | L | S |

Table 23: Best hyperparameter for each model used in the Airfoil experiments.

| Model | Hyperparameter Name | Operator Learining |
|---|---|---|
| Ours | Attention layers number | 16 |
| | Attention hidden dimension | 64 |
| | Attention head number | 5 |
| | FFE dimension | 2 |
| | AGF depth | 15 |
| | En/Decoder hidden dimension | 512 |
| DeepONet | Branch net depth | 6 |
| | Trunk net depth | 5 |
| | Hidden dimension | 256 |
| Oformer | Layers number | 3 |
| | Hidden dimension | 64 |

