# OpenReview forum: "PDE-PFN: Prior-Data Fitted Neural PDE Solver"
_ICLR.cc/2026/Conference — Submitted to ICLR 2026_

### Official Review · Reviewer_7z9q · 2025-10-30

**Soundness:** 3
**Presentation:** 3
**Contribution:** 3
**Rating:** 6
**Confidence:** 4

**Summary:**

This paper proposed to use prior-data fitted networks (PFNs) which uses a Transformer to perform in-context Bayesian inference to approximate the posterior predictive distribution (PPD) of PDE solutions. Prior knowledge is from the predictions of PINNs. The in-context correction is when solving a new PDE, accurate simulated known boundary conditions and initial conditions data will be provided for in-context learning by the Transformer. The main contribution is the robustness to noisy priors and strong generalization to different PDEs and the ability of Zero-shot inference. Comparing to current methods, this method does not require explicit governing PDE which increases the flexibility of data using.

**Strengths:**

This paper has high level of novelty in using PFNs and ICL and improved the generalizability of Neural PDE solver to a high level.
This paper has a rich baseline models to compare with and outperformed all of the baselines which is a pretty strong empirical result.

**Weaknesses:**

The only question I have is this: PINNs are known to have failure modes when the PDE coefficient is large, causing ill-conditioning and very complex loss landscapes. I notice that the PDEs tested in this paper generally have low coefficients. My concern is that when the prior is based on PINNs solving high-coefficient PDEs, the prior data would be fundamentally wrong (i.e., highly inaccurate). Given this situation, how well can the ICL perform in accurately predicting the solution to new PDEs, especially when the new PDE also has a high coefficient?

**Questions:**

None.

**Details Of Ethics Concerns:**

None.

---

> ### Author Response · Authors · 2025-11-21
> **Response to reviewer 7z9q**
>
> ## **Response to reviewer 7z9q**
> Thank you very much for your positive evaluation of our work. We sincerely appreciate your thoughtful comments and have prepared detailed responses to each of your questions. We hope that our clarifications help address your concerns. If further clarification or additional experiments would be helpful, we would be happy to provide them.
>
> In addition, during the rebuttal period we conduct several new experiments motivated by questions from all reviewers. These results strengthen the empirical validation of our method and further clarify its contributions. We have incorporated all newly added or fixed contents into the revised manuscript, with changes highlighted in blue for clarity.
>
> The new experiments are summarized as follows:
>
> - **Main paper Section 4** : Additional baseline - Latent Neural operator(LNO)
> - **Appendix H.5** : Unseen coefficient unseen temporal domain 2D CDR equation experiment
> - **Appendix H.6** : 3D diffusion-reaction equation experiment
> - **Appendix H.7** : Ablation study
> - **Appendix H.8** : Biased-noise robustness experiment
> - **Appendix H.9** : Computational cost analysis
>
> ### **[Weakness]**
> Thank you for the thoughtful question. We fully agree that *vanilla* PINNs can experience severe failure modes when PDE coefficients are large, due to ill-conditioned optimization and complex loss landscapes. However, our use of PINN priors is not based on the assumption that *vanilla* PINNs can reliably solve such regimes. Instead, they serve as a noisy prior, reflecting the type of low-cost approximations that frequently arise in real-world scientific workflows. Our primary goal is to evaluate the robustness of PDE-PFN to noisy priors using PINN prior.
>
> In practice, obtaining a rough surrogate, possibly inaccurate or biased, is often far more feasible than obtaining a fully converged high-fidelity solution. The PINN prior in our experiments intentionally serves as an example of such a noisy prior. We emphasize that our method does not depend on PINNs specifically. If higher-quality or more efficient solutions are available, such as coarse numerical solvers or more advanced PINN variants such as Piratenets [1].
>
> Importantly, the Bayesian in-context learning formulation of PDE-PFN is designed to remain stable even when the prior is noisy. We confirm the robustness of our model when using PINN priors. Moreover we conduct additional experiment to further validate this robustness by injecting biased Gaussian noise to numerical prior during the inference stage. Detailed results can be found in our **response to Question 3**.
>
> [1] Wang, Sifan, et al. "Piratenets: Physics-informed deep learning with residual adaptive networks." Journal of Machine Learning Research 25.402 (2024): 1-51.

---

> > ### Comment · Reviewer_7z9q · 2025-11-25
> >
> > Thank you for your response. Most of my concerns have been resolved. I recommend that the authors include the additional experiments in the revised paper.

---

> > > ### Author Response · Authors · 2025-11-26
> > > **Response to reviewer 7z9q**
> > >
> > > Thank you once again for your thoughtful follow-up and for taking the time to review our rebuttal. We are very glad to hear that most of your concerns have been resolved. We will carefully incorporate the additional experiments and clarifications into the revised version. If there are any remaining questions or clarifications you would like to discuss, we would be more than happy to address them.
> > >
> > > We hope that the additional experiments and clarifications provided offer a clearer view of the contributions and strengths of our work, and we would be grateful if you take them into consideration during the final assessment.
> > >
> > > Thank you again for your careful evaluation and your contribution to the review process.

---

### Official Review · Reviewer_HzTi · 2025-10-30

**Soundness:** 2
**Presentation:** 3
**Contribution:** 2
**Rating:** 4
**Confidence:** 3

**Summary:**

The paper presents a novel architecture called PDE-PFN based on the Prior-data fitted network that, given a set of spatial samples from a trajectory, can predict the rest of the solution. This can include interpolation as well as extrapolation tasks. The method can be seen as in-context Bayesian inference. The base model is a transformer with attentive graph filter layers, and the input coordinates are encoded using a Fourier embedding. The method is evaluated mainly on convection-diffusion-reaction tasks, as well as Navier-Stokes and shallow-water equations.

**Strengths:**

1. Interesting model architecture that can perform solution interpolation and temporal extrapolation at the same time.
2. Introduces theory from prior-fitted neural networks to PDEs.

**Weaknesses:**

1. The paper aims to overcome the limitations of neural PDE solvers to deal with irregular meshes. However, there are many methods that can deal with such domains [1,2,4].
2. The framing of the setup as "in-context learning" is a bit confusing. The tasks the model generalizes between are actually just solutions of different settings of the PDE (i.e., different ICs for the later experiments). Each task here refers to a single PDE solution/trajectory. In this sense, all neural operators that predict the solution given the IC would be in-context learners. In the original PFN paper, the datasets used in training are each independent tabular datasets, for example. The closest translation to the PDE domain would be to have a model that takes in different trajectories for a given target PDE parameter as the context, and uses this context to infer the solution for another IC of the same parameter. The setup presented in the paper is more akin to masked-pretraining strategies [3], where parts of the trajectory are masked out and the model is trained to fill in the blanks.
3. Only FNO-variants and two PDE foundation models are chosen as the baselines. Since the presented model is a transformer variant, it would be good to evaluate against other recent transformer models such as [1,2,4].
4. The experiments don't show how much the model actually profits from the ICL formulation. For example, one experiment to do would be to look at the unseen parameters in the temporal extrapolation case. Here, one could use the current architecture, only train with the input points at the first time step, and points on the solution trajectory as the query points. This would be the same setup as LNO, for example.

[1] Luo, H., Wu, H., Zhou, H., Xing, L., Di, Y., Wang, J., & Long, M. (2025). Transolver++: An Accurate Neural Solver for PDEs on Million-Scale Geometries.
[2] Wang, T., & Wang, C. (2024). Latent neural operator for solving forward and inverse pde problems.
[3] Zhou, A., & Farimani, A. B. (2024). Masked autoencoders are PDE learners.
[4] Serrano, L., Wang, T. X., Le Naour, E., Vittaut, J. N., & Gallinari, P. (2024). AROMA: Preserving spatial structure for latent PDE modeling with local neural fields.

**Questions:**

1. How are the other models conditioned on the PDE parameters in 4.2?
2. During temporal extrapolation: The context consists only of all the spatial points at t=0.1, right? Why did you not start at t=0?
3. What are the initial conditions in the CDR equations?
4. What exactly do you mean by "minimal fine-tuning? What happens if you also use, for example, the pretrained FNO from the first task and fine-tune it on 4.3? Does the PDE-PFN profit more from being pretrained than the other models?

---

> ### Author Response · Authors · 2025-11-21
> **Response1 to reviewer HzTi**
>
> ## **Response to reviewer HzTi**
> Thank you for the valuable feedback on our paper. We hope that our responses will address the reviewer’s questions effectively. We have incorporated the additional elements discussed during the rebuttal process into the revised paper, and all newly added or fixed content is highlighted in blue text. If further clarification or additional experiments would be helpful, we would be happy to provide them.
>
> ### **[Weakness 1]**
> Thank you for pointing this out. We fully agree that handling irregular meshes is no longer unique, and recent methods such as Latent Neural Operators (LNO)[1] have indeed demonstrated strong performance on irregular domains. Our intention was not to claim novelty solely in irregular-domain compatibility. Rather, irregular-mesh compatibility is only one component of the broader contribution of PDE-PFN. Our work primarily aims to demonstrate that a PFN-based in-context learning framework can support: Task generalization without task-specific re-training, PDE generalization across heterogeneous equations with minimal fine-tuning, Robustness to low-cost and noisy priors, and Flexible input/output structure including irregular meshes.
>
> While existing neural operator approaches can process irregular geometries, they typically learn a single fixed operator and require retraining whenever the underlying PDE, the boundary conditions, or even the task formulation changes. In contrast, our model uses in-context Bayesian inference to approximate the posterior predictive distribution, enabling it to adapt to new tasks in a unified way.
>
> [1] Wang, T., & Wang, C. (2024). Latent neural operator for solving forward and inverse pde problems.
>
> ### **[Weakness 2]**
>
> Thank you for raising this point. Our use of the term in-context learning follows the PFN formulation, where the model performs Bayesian inference conditioned on arbitrary context sets, rather than learning a single fixed operator.
>
>
> For example, the model may be trained to predict the solution at $t=0.5$ with contexts at $t=0.0$, but evaluated to predict the solution at $t=1.0$ using contexts at $t=0.25$. This property is not supported by standard neural operators, which learn a fixed operator (e.g., $IC → u(\cdot, T)$) and require the input-output structure during training and testing to match exactly. Because PDE-PFN does not bind itself to a fixed operator during training, it can generalize across different tasks without task-specific fine-tuning.
>
> We acknowledge that masked-pretraining approaches also mask trajectory points, but their objective is to learn a single PDE operator, not to perform inference across diverse conditions and tasks. In contrast, our method conditions on the provided arbitrary context as an implicit prior and performs Bayesian inference.
>
> We further explain the distinctions between our approach and other in-context learning-based neural operators in our **response to Reviewer k7Hr’s Weakness 2**.
>
> ### **[Weakness 3]**
>
> Thank you for the suggestion. Among the transformer-based models you mentioned, we implement the Latent Neural Operator (LNO) and evaluate it on the same set of experiments presented in the main paper. We search over the hyperparameters reported as optimal across the datasets in the original LNO paper and select the configuration that achieves the best validation performance in our setting.
>
> | task | Metric | 2D CDR equations |  |  | SWE | |CNSE |
> |-|-|-|-|-|-|-|-|
> | | | Temporal extrapolation | Coefficient interpolation | Coefficient extrapolation | Seen | Unseen | Operator Learning |
> | Ours | $L_1$ abs | 0.01474 | 0.08957 | 0.01027 | 0.01548 | 0.01571 | 0.09497 |
> | | $L_2$ rel | 0.02261 | 0.11990 | 0.01068 | 0.03273 | 0.03323 | 0.54833 |
> | | $L_{\infty}$ rel | 0.08444 | 0.29526 | 0.01661 | 0.20590 | 0.20686 | 0.63533 |
> | LNO | $L_1$ abs | 0.02876 | 0.11085| 0.05076 | 0.02902 | 0.02877 | 0.10366 |
> | | $L_2$ rel | 0.03665 | 0.14913| 0.06347 | 0.05269 | 0.05225 | 0.62536 |
> | | $L_{\infty}$ rel | 0.10948 | 0.33280 | 0.15091 | 0.24832 | 0.24788 | 0.66936 |
>
> Experimental results show that PDE-PFN outperforms LNO across all tasks. We have added these results to the experimental section (Section 4) of the revised manuscript.

---

> ### Author Response · Authors · 2025-11-21
> **Response2 to reviewer HzTi**
>
> ### **[Weakness 4]**
>
> In our original coefficient interpolation/extrapolation experiments, we intentionally kept the temporal test domain aligned with the training domain in order to isolate the effect of varying the PDE coefficients. This was meant to ensure that the evaluation reflected parameter generalization rather than a mixture of parameter and temporal-domain shifts. However, we agree that evaluating the model under setup similar to LNO provides an additional perspective on the effectiveness of the in-context formulation.
>
> Following your recommendation, we conduct an experiment that directly mirrors the setup: for unseen coefficient settings, the model is given only the solution at a single time step at $t=0.6$ and is required to infer the subsequent trajectory at $t=0.7,0.8,0.9,$ and $1.0$.
>
> | Task | Metric | Ours | Ours (PINN) | DeepONet | FNO | F-FNO | A-FNO | LNO | Poseidon | DPOT |
> |-|-|-|-|-|-|-|-|-|-|-|
> | Inter Coeff | $L_1$ abs | 0.00736 | 0.00939 | 0.05139 | 0.10863 | 0.01405 | 0.04695 | 0.01839 | 0.13852 | 0.04610 |
> |  | $L_2$ rel | 0.00954 | 0.01178 | 0.06179 | 0.14380 | 0.01636 | 0.06227 | 0.02049 | 0.14993 | 0.04676 |
> |  | $L_{\infty}$ rel | 0.03193 | 0.03693 | 0.16008 | 0.34858 | 0.03847 | 0.21750 | 0.04512 | 0.26293 | 0.04676 |
> | Extra Coeff | $L_1$ abs | 0.01027 | 0.02063 | 0.05514 | 0.03887 | 0.02312 | 0.05013 | 0.02560 | 0.14661 | 0.05186 |
> |  | $L_2$ rel | 0.01068 | 0.02169 | 0.06450 | 0.05114 | 0.02470 | 0.06351 | 0.02677 | 0.15650 | 0.05189 |
> |  | $L_{\infty}$ rel | 0.01661 | 0.03474 | 0.16555 | 0.13713 | 0.04591 | 0.20955 | 0.04911 | 0.27502 | 0.05189 |
>
> Both versions of PDE-PFN outperform all baselines, demonstrating that the model is capable of inferencing on unseen coefficients and unseen temporal domains using only the provided contexts based on ICL.
>
> We have included these new results to **Appendix H.5** of the revised manuscript.
>
>
> ### **[Question 1]**
>
> In all experiments in Section 4.2, we ensure that every baseline model was conditioned in exactly the same way as PDE-PFN. Since PDE-PFN receives only the spatial–temporal coordinates and corresponding solution values of the context points, we provided the identical set of context coordinates and solution samples to all baseline models as well. No additional information such as the governing PDE, symbolic form, or its coefficients was supplied to any model.
>
> Therefore, all models were required to infer the future trajectory purely from the contexts, without access to any PDE-specific metadata. This setup guarantees a fully controlled and fair comparison across architectures, since each model receives the same conditioning inputs and must solve the same inference problem.
>
>
> ### **[Question 2]**
>
> Thank you for the question. As we further explain in our **response to Question 3**, all CDR equations in our dataset share the same initial condition at $t=0.0$. Therefore, if we used $t=0.0$ as the only context, every model, including the baselines, would receive identical inputs for every PDE, regardless of the underlying coefficients. Under such a setup, no model could distinguish between different PDE parameter settings, and the extrapolation task would become ill-posed.
>
> By contrast, at $t=0.1$, the solutions have already diverged according to their respective coefficients. Using the field at $t=0.1$ as the context allows all models to condition on meaningful information that actually reflects the underlying PDE parameters. This ensures a fair and informative temporal extrapolation setup.
>
> ### **[Question 3]**
>
> All CDR equations were generated with the same initial condition, $u(x,0)=1+sin(2x)sin(2y)$. We intentionally fixed the initial condition for all equations because varying both the PDE coefficients and the initial conditions simultaneously would make it impossible for any models to disentangle whether differences in the solution trajectory arise from changes in the PDE coefficients or from changes in the initial conditions.
>
> To ensure that the evaluation truly reflects parameter generalization rather than mixed effects, we varied only the coefficients while keeping the initial condition fixed for all CDR instances.

---

> ### Author Response · Authors · 2025-11-21
> **Response3 to reviewer HzTi**
>
> ### **[Question 4]**
>
> We mean that PDE-PFN was fine-tuned using only half the number of epochs used for training baselines which is not foundation model. This reduced schedule was chosen to demonstrate that the pretrained model can adapt to new PDEs efficiently, without relying on extensive retraining.
>
> Following your suggestion, we also evaluate whether a pretrained neural operator baseline would benefit similarly. Specifically, we take the best-performing FNO configuration from the 2D CDR experiments and fine-tune it on CNSE using the same reduced fine-tuning budget as PDE-PFN. The results are summarized below:
>
> ||Ours|FNO trained from scratch|FNO with fine-tuning|
> |-|-|-|-|
> |$L_1$ abs error |0.09497|0.09581|0.09854|
> |$L_2$ rel error |0.54833|0.57905|0.71421|
> |$L_\infty$ rel error |0.63533|0.70708|0.81866|
>
> We find that FNO, when limited to this minimal fine-tuning regime, actually perform worse than training from scratch with its full training schedule. In contrast, PDE-PFN benefits from pretraining and retains strong performance even under this restricted fine-tuning budget.
>
> This indicates that the cross-PDE generalization of PDE-PFN is not merely due to fine-tuning, but stems from the underlying PFN-based in-context learning mechanism, which allows the model to transfer information across PDE families more effectively than neural operator baselines.

---

> > ### Comment · Reviewer_HzTi · 2025-11-24
> >
> > Thank you for your response. I have a few follow-up questions:
> >
> > 1. Is the task unique in the sense that a rollout of the IC for different PDE parameters gives us only different states?
> > 2. Would the approach still work when we have also different initial conditions? I understand now that your model generalizes to new physics since every trajectory in your pre-training data set comes from a different parameter with the same IC.
> > 3. How did you train the operator baselines? For example, with full autoregressive rollouts or 1-step predictions?

---

> > > ### Author Response · Authors · 2025-11-25
> > > **Response to follow-up question of reviewer HzTi**
> > >
> > > ## **Response to follow-up question of reviewer HzTi**
> > > Thank you for your constructive follow-up question. We hope the following clarification resolves your concerns. All experiments conducted for this response have been added to the revised paper, with new content highlighted in blue.
> > >
> > > ### **[Question 1]**
> > >
> > > Thank you for the thoughtful question. It is theoretically possible for two different CDR equations to produce the same state at a specific time point. However, it is very rare that different PDE parameters generate the same state during a long time period. Therefore, if a sufficiently long context is provided, its uniquely determine solution can be identified and according to Theorem 2.1, the correctness of our inference outcome is also guranteed.
> > >
> > > In addition, we also conduct the sanity check for our dataset. We examin all contexts used in our experiments and found no cases where two distinct context shared exactly the same spatiotemporal coordinates and identical solution values. This is the case not only for the CDR dataset but also for the SWE and CNSE experiments presented in the main paper. Therefore, in all tasks we considered, each trajectory is uniquely different from others, ensuring that the model receives distinguishable context signals.
> > >
> > >
> > > ### **[Question 2]**
> > > Thank you for the constructive question. To directly evaluate whether PDE-PFN continues to work under different initial condition, we conduct an additional experiment following your suggestion. Specifically, we replace the original initial condition with a new one, $u\_0(x,y)=1+sin(3x)sin(3y)$, and train the model on this modified dataset. The results are summarized below:
> > >
> > > | Task | Metric | Ours | DeepONet | FNO | F-FNO | A-FNO | LNO | Poseidon | DPOT |
> > > |-|-|-|-|-|-|-|-|-|-|
> > > | **Spatiotemporal interpolation** | $L\_1$ abs | 0.01339 | 0.07394 |  |  |  |  |  |  |
> > > | | $L\_2$ rel | 0.01885 | 0.09376 |  |  |  |  |  |  |
> > > | | $L\_{\infty}$ rel | 0.06606 | 0.26320 |  |  |  |  |  |  |
> > > | **Temporal extrapolation** | $L\_1$ abs | 0.02017 | 0.03988 | 0.03184 | 0.03371 | 0.08342 | 0.04696 | 0.09397| 0.12628 |
> > > | | $L\_2$ rel | 0.02929 | 0.05282 | 0.04160 | 0.04192 | 0.09967 | 0.06282 | 0.10352 | 0.13633 |
> > > | | $L\_{\infty}$ rel | 0.10793 | 0.17487 | 0.12671 | 0.10980 | 0.27716 | 0.18313 | 0.23855 | 0.13633 |
> > >
> > > Across both tasks, PDE-PFN continues to achieve the best performance, even when trained on different initial condition. This confirms that the method does not rely on a specific IC structure. Moreover, the experiments on SWE and CNSE in the main paper already involve datasets where the same underlying PDE is paired with many different initial conditions sampled from a distribution. The strong performance reported there also indicates that PDE-PFN is not restricted to scenarios where all trajectories originate from a single fixed IC.
> > >
> > > We have added this new experiment to Appendix H.10 in the revised manuscript.
> > >
> > >
> > > ### **[Question 3]**
> > > All operator baselines were trained using 1-step prediction. Although PDE-PFN is capable of modeling multi-step trajectories in a single forward pass, we trained it under the same 1-step setting for fairness. This ensures that all models were optimized under identical prediction horizons and supervision.

---

> ### Author Response · Authors · 2025-11-27
> **Reminder for reviewer HzTi**
>
> ## **Reminder for reviewer HzTi**
>
> Thank you very much for your thoughtful and constructive feedback on our submission. We sincerely appreciate the time you have taken to evaluate our work. We have carefully addressed all comments and prepared detailed responses to each question. We hope that our clarifications fully resolve your concerns, and we would be happy to provide further explanations or additional experiments if needed.
>
> During the rebuttal period, we also conducted new experiments inspired by the reviewers’ questions, which further strengthen the empirical validation of our method. All newly added or revised content has been incorporated into the updated manuscript, with changes highlighted in blue for clarity.
>
> A summary of the newly added experiments is as follows:
>
> - **Main paper Section 4** : Additional baseline - Latent Neural operator(LNO)
> - **Appendix H.5** : Unseen coefficient unseen temporal domain 2D CDR equation experiment
> - **Appendix H.6** : 3D diffusion-reaction equation experiment
> - **Appendix H.7** : Ablation study
> - **Appendix H.8** : Biased-noise robustness experiment
> - **Appendix H.9** : Computational cost analysis
> - **Appendix H.10** : 2D CDR equation experiment with different initial condition
>
> Thank you again for your time and effort in reviewing our paper. Please feel free to let us know if you have any additional questions. We would be glad to assist.

---

> > ### Comment · Reviewer_HzTi · 2025-11-27
> >
> > Thank you for your clarifications and experiments. I have increased my score. However, I still want to add that it would be beneficial to include an explanation of the single initial condition in the main paper, as this is a rather unusual setting.

---

> > > ### Author Response · Authors · 2025-11-27
> > > **Response to reviewer HzTi**
> > >
> > > Thank you very much for your positive evaluation and for taking the time to re-examine our rebuttal. We sincerely appreciate your thoughtful feedback and the increase in your score.
> > >
> > > As you suggested, we have updated the main paper to explicitly clarify that the family of 2D CDR equations was generated using a single shared initial condition. This revision has been incorporated into Section 4.1, and the corresponding sentence has been revised accordingly in the updated manuscript (highlighted in blue).
> > >
> > > Thank you again for your constructive comments and for helping us improve the clarity of the paper.

---

### Official Review · Reviewer_AmpV · 2025-11-01

**Soundness:** 3
**Presentation:** 2
**Contribution:** 2
**Rating:** 6
**Confidence:** 4

**Summary:**

This paper introduces PDE-PFN, a prior-data fitted neural network framework for solving partial differential equations via direct approximation of the posterior predictive distribution using in-context Bayesian inference. The proposed method extends the PFN architecture with specialized architectural choices including Fourier feature embeddings and attentive graph filter Transformers. The experimental results demonstrate flexibility in input formats, robustness to noisy priors, zero-shot inference, and improved or competitive performance against state-of-the-art neural PDE solvers and scientific foundation models.

**Strengths:**

1.	PDE-PFN demonstrates strong generalization capacities across both parameterized PDE families and heterogeneous PDEs (e.g., SWE, CNSE), outperforming or matching established task-specific and foundation model baselines.
2.	The method’s ability to provide stable and high-quality solutions even when pre-trained on approximated, noisy PINN priors is convincing
3.	Unlike neural operators constrained to grid data or fixed points, PDE-PFN supports mesh, irregular, and arbitrary-coordinate input/output configurations, as systematically compared in Table 7

**Weaknesses:**

1.	Although the manuscript is explicit that extension to higher-dimensional or more complex PDEs is left to future work. The scope is currently limited to simple 2D systems. The generalizability and scalability to realistic 3D scientific settings or highly nonlinear, multi-physics PDEs remain untested, detracting from the claimed universality.
2.	While Fourier features, AGF layers, and the Transformer modifications are pitched as improvements, their individual impact isn’t adequately disentangled. For instance, no ablation is provided isolating the benefit (or necessity) of the AGF layer over, say, vanilla Transformer blocks or simple attention.
3.	Equation (3) defines an MSE on PINN priors, but there is little elaboration on the sampling strategies, distribution of priors, or potential class-imbalance effects if the PDE parameter space is highly inhomogeneous.

**Questions:**

1.	Can the authors provide an ablation study that isolates the incremental impact of AGF layers, Fourier feature embedding, and rational activation on generalization? How would vanilla Transformer, or sinusoidal vs. rational activation, affect baseline and robustness metrics?
2.	The evaluation is mainly on 2D PDEs. Have the authors attempted extending PDE-PFN to 3D cases, higher spatial/temporal resolutions, or PDEs with complex/mixed boundary conditions? If not, what do the authors see as key obstacles?
3.	For the regularization term on the AGF layer, does its influence on overall loss vary greatly with $N_D$ or network depth? How sensitive are results to this regularizer's weighting?

---

> ### Author Response · Authors · 2025-11-21
> **Response1 to reviewer AmpV**
>
> ## **Response to reviewer AmpV**
> Thanks for your valuable feedback. We hope that our responses will address the reviewer’s questions effectively. We have incorporated the additional elements discussed during the rebuttal process into the revised paper, and all newly added or fixed content is highlighted in blue text. If further clarification or additional experiments would be helpful, we would be happy to provide them.
>
> ### **[Weakness 1]**
> Thank you for this constructive suggestion. In response, we generate a new dataset that could be produced within the short rebuttal period to directly assess scalability beyond 2D. Specifically, we construct a new 3D diffusion–reaction equation dataset by extending the exact same procedure used for generating the 2D CDR equations. This produces three diffusion coefficients $(\nu_x, \nu_y, \nu_z)$ and one reaction coefficient $\rho$, each sampled from $\{0.0, 0.5, 1.0,1.5\}$, yielding in 256 distinct PDE instances.
>
> Using this dataset, we evaluated the same two tasks as in the 2D CDR experiments: spatiotemporal interpolation and temporal extrapolation. Among our baselines, only DeepONet, FNO, and DPOT support 3D data. thus, we compare ours against these models. For our model and DPOT, we perform fine-tuning, using only half the number of epochs employed for the other baselines. The results are shown below:
>
> | Task | Metric | Ours | Ours (PINN) | DeepONet | FNO | DPOT |
> |-|-|-|-|-|-|-|
> | **Spatiotemporal interpolation** | $L_1$ abs | 0.02685 | 0.02843 |  0.02816 |  |  |
> | | $L_2$ rel | 0.03711 | 0.04144 | 0.03981 |  |  |
> | | $L_{\infty}$ rel | 0.10938 | 0.12823 | 0.12724 |  |  |
> | **Temporal extrapolation** | $L_1$ abs | 0.01031 | 0.01039 | 0.02421 | 0.01521 | 0.01485 |
> | | $L_2$ rel | 0.01376 | 0.01391 | 0.03033 | 0.01824 | 0.01638 |
> | | $L_{\infty}$ rel | 0.04180 | 0.04545 | 0.11530 | 0.05545 | 0.01638 |
>
>
> Across both tasks, our model outperforms the baselines in spatiotemporal interpolation in all metrics and in temporal extrapolation on two metrics, demonstrating scalability to 3D PDEs even with minimal fine-tuning. While we view these experiments as preliminary rather than definitive evidence of full “universality” across all 3D or multi-physics PDEs, they provide concrete support that PDE-PFN’s generalization capabilities are not restricted to 2D domains. We have included these new results in **Appendix H.6** of the revised paper.
>
> ### **[Weakness 2]**
>
> Thank you for suggesting a deeper ablation analysis. In response, we perform ablation studies on two major components of our model: AGF[1] layers, and the rational activation. As noted in Appendix I.3, when the input feature dimensionality changes across datasets (e.g., 2D CDR and SWE vs. CNSE), the encoder start and decoder end layers must be re-initialized. As a result, the optimal Fourier feature configuration differs across datasets, and for the 2D CDR experiments, the best hyperparameters actually did not use Fourier features. Therefore, we focus our analysis on the components that remain consistent across datasets: AGF and rational activations.
>
> Below are the results. “Ours w.o. agf”, “Ours w.o. act”, and “Ours w.o. all” denote replacing AGF with a *vanilla* Transformer block, replacing the rational activation with GELU, and replacing both, respectively.
>
> | Task | Metric | Ours | Ours w.o. agf| Ours w.o. act | Ours w.o. all |
> |-|-|-|-|-|-|
> | **Spatiotemporal interpolation** | $L_1$ abs | 0.01218  | 0.01242 | 0.01040 | 0.01780 |
> | | $L_2$ rel | 0.01643 | 0.01625 | 0.01495 | 0.02281 |
> | | $L_{\infty}$ rel | 0.04600 | 0.04297 | 0.04399 | 0.05736 |
> | **Temporal extrapolation** | $L_1$ abs | 0.01474 | 0.01776 | 0.01725 | 0.02454 |
> | | $L_2$ rel | 0.02261 | 0.02779 | 0.02665 | 0.03437 |
> | | $L_{\infty}$ rel | 0.08444 | 0.09554 | 0.09376 | 0.10625 |
>
> The results show that, although some ablated variants achieve comparable or slightly improved accuracy on the spatiotemporal interpolation task, our full model consistently outperforms all ablations on the temporal extrapolation task, which requires stronger out-of-distribution generalization. This supports our claim that the proposed architectural choices improve the model’s ability to generalize across diverse tasks in a stable and efficient manner. As also confirmed in the **response to Question 1**, our model demonstrate superior performance in terms of robustness to noise. We have included these new results to **Appendix H.7** of the revised manuscript.
>
> [1]  Wi, Hyowon, Jeongwhan Choi, and Noseong Park. (2025). Learning Advanced Self-Attention for Linear Transformers in the Singular Value Domain.

---

> ### Author Response · Authors · 2025-11-21
> **Response2 to reviewer AmpV**
>
> ### **[Weakness 3]**
>
> Thank you for pointing this out. In the main paper, Equation (3) was written in a unified form because fine-tuning datasets such as SWE and CNSE do not admit a coefficient–dictionary representation. Instead, they consist of samples generated from the single PDE under different initial conditions. In contrast, the 2D CDR pretraining dataset is generated from a coefficient dictionary, allowing a parameter-space description. To present an objective that is valid for both settings, we expressed Eq.(3) in a more general form.
>
> During pretraining, each of the 2,187 CDR PDE equation is used exactly once per epoch, and for each equation, spatial–temporal query points in Eq. (3) are drawn identically over the equations. Thus, the parameter space is sampled uniformly, and every equation contributes equally to the loss. No region of the coefficient space is overrepresented, and class imbalance does not occur. For datasets like SWE and CNSE, where no explicit parameter dictionary exists, all training instances are drawn uniformly from the same underlying PDE, so imbalance is not a concern.
>
>
> ### **[Question 1]**
>
> Thank you for this productive question. The primary ablation results for each architectural component are provided in our **response to Weakness 2**. To further evaluate robustness, particularly under noisy priors, we conduct an additional experiment using PINN prior. Specifically, we compare our full model against two variants: “Ours w.o. agf,” which replaces the AGF layer with a *vanilla* Transformer block, and “Ours with sinusoidal,” which replaces the rational activation with a standard sinusoidal activation. The results are summarized below:
>
> | Task | Metric | Ours | Ours w.o. agf| Ours with sinusoidal |
> |-|-|-|-|-|
> | **Spatiotemporal interpolation** | $L_1$ abs | 0.01507  | 0.01667 | 0.01482 |
> | | $L_2$ rel | 0.02057 | 0.02171 | 0.02096 |
> | | $L_{\infty}$ rel | 0.06584 | 0.05536 | 0.06946 |
> | **Temporal extrapolation** | $L_1$ abs | 0.01940 | 0.02105 | 0.02671 |
> | | $L_2$ rel | 0.02705 | 0.03030 | 0.04159 |
> | | $L_{\infty}$ rel | 0.08802 | 0.09518 | 0.14922 |
>
> Across both tasks, our full model consistently demonstrates stronger robustness than the ablated variants, particularly in the temporal extrapolation setting. These findings reinforce the importance of the AGF layer and rational activation for robustness of the model. We have added these new results to **Appendix H.7** of the revised manuscript, together with the earlier ablation studies.
>
> ### **[Question 2]**
>
> Thank you for the insightful question. We have indeed explored extending PDE-PFN beyond 2D within the rebuttal period. Specifically, we conduct experiments on a 3D diffusion–reaction equation. Details can be found in the **response of Weakness 1**.
>
> ### **[Question 3]**
>
> Thank you for the progressive question. As noted, the AGF regularization term is averaged across all layers, so increasing the number of layers affects primarily the rate at which the term converges during early training, not the final magnitude. To evaluate the sensitivity to the weighting factor, we remove the scaling term and retrained the model under identical settings.
>
> | Task | Metric | Ours | Ours w.o. agf loss scaling|
> |-|-|-|-|
> | **Spatiotemporal interpolation** | $L_1$ abs | 0.01507  | 0.02739 |
> | | $L_2$ rel | 0.02057 | 0.03548 |
> | | $L_{\infty}$ rel | 0.06584 | 0.08414 |
> | **Temporal extrapolation** | $L_1$ abs | 0.01940 | 0.02895 |
> | | $L_2$ rel | 0.02705 | 0.04511 |
> | | $L_{\infty}$ rel | 0.08802 | 0.15917 |
>
>
> Removing the scaling factor lead to noticeably worse performance. To better understand why, we examine the relative contribution of the AGF loss during early training epochs.
>
> | Epoch | Ours | Ours w.o. agf loss scaling|
> |-|-|-|
> | Before training (first iteration) | 0.032% | 24.733% |
> | After First epoch  | 0.973% | 90.768% |
> | After Second epoch | 1.843% | 94.084% |
> | After Third epoch  | 3.809% | 97.291% |
> | After Fourth epoch | 4.164% | 97.871% |
> | After Fifth epoch  | 6.136% | 98.419% |
>
>
> The measurements reveal that without scaling, the AGF loss dominate more than 90% of the total training loss after the first epoch, overwhelming the prediction loss. This imbalance prevents the model from effectively optimizing the predictive objective, confirming that the scaling factor is essential for stabilizing training.
> With the scaling in place, the AGF loss remains properly regularized and does not interfere with the main learning signal. This confirms that the weighting is essential for maintaining stable optimization, while the averaging over layers ensures that depth does not disproportionally amplify the regularization term.

---

> ### Author Response · Authors · 2025-11-27
> **Reminder for reviewer AmpV**
>
> ## **Reminder for reviewer AmpV**
>
> Thank you very much for your thoughtful and constructive feedback on our submission. We sincerely appreciate the time you have taken to evaluate our work. We have carefully addressed all comments and prepared detailed responses to each question. We hope that our clarifications fully resolve your concerns, and we would be happy to provide further explanations or additional experiments if needed.
>
> During the rebuttal period, we also conducted new experiments inspired by the reviewers’ questions, which further strengthen the empirical validation of our method. All newly added or revised content has been incorporated into the updated manuscript, with changes highlighted in blue for clarity.
>
> A summary of the newly added experiments is as follows:
>
> - **Main paper Section 4** : Additional baseline - Latent Neural operator(LNO)
> - **Appendix H.5** : Unseen coefficient unseen temporal domain 2D CDR equation experiment
> - **Appendix H.6** : 3D diffusion-reaction equation experiment
> - **Appendix H.7** : Ablation study
> - **Appendix H.8** : Biased-noise robustness experiment
> - **Appendix H.9** : Computational cost analysis
> - **Appendix H.10** : 2D CDR equation experiment with different initial condition
>
> Thank you again for your time and effort in reviewing our paper. Please feel free to let us know if you have any additional questions. We would be glad to assist.

---

### Official Review · Reviewer_k7Hr · 2025-11-01

**Soundness:** 2
**Presentation:** 3
**Contribution:** 2
**Rating:** 6
**Confidence:** 4

**Summary:**

The paper proposes PDE-PFN, a prior-data fitted neural solver that uses a PFN-style Transformer with self- and cross-attention to approximate the posterior predictive distribution of PDE solutions via in-context learning. The model is pre-trained on a broad “family” of 2D convection–diffusion–reaction equations.

**Strengths:**

1. Methodological novelty and coherence: importing PFN/ICL ideas to PDE solving and training on noisy PINN priors as "cheap" prior data is conceptually neat, practical, and aligns with the Bayesian predictive distribution perspective.
2. Input flexibility: ability to take arbitrary context/query sets (including irregular meshes) is important in real-world sensor-driven settings; the Airfoil and Darcy experiments substantiate this.
3. Robustness to noisy priors: training with PINN-based approximations yet surpassing baselines indicates the proposed Bayesian-ICL mechanism is effective under noisy pretraining distributions.

**Weaknesses:**

1. While the paper is upfront about this, claims toward “foundation” capabilities would be more compelling with complex PDEs (Navier–Stokes) or complex geometries with strong stiffness.
2. PINNs are not universally cheap or stable; their training cost and failure modes vary across PDEs/BCs/ICs. The paper could better quantify the end-to-end cost trade-off (pretraining PINNs for 2187 CDR variants is nontrivial).
3. Positioning vs. recent in-context operator learning SFMs (e.g., ICON-LM, OmniArch) is unclear. The paper claims zero-shot ICL without demos and emphasizes Bayesian-flavored PPD approximation, but it does not clearly articulate architectural and training differences relative to these closely related lines, nor why those differences matter empirically.

**Questions:**

1. What is the total computational budget for generating the 2187 PINN priors and training PDE-PFN versus training a strong neural operator baseline on the same tasks?
2. How does training and inference scale with (i) number of context points, (ii) query points, and (iii) spatial/temporal resolution? Any memory/time complexity analysis for AGF layers in this setting?
3. Your theorem addresses zero-mean noise. PINN errors can be biased. How sensitive is PDE-PFN to biased priors? Can you simulate controlled bias in priors and report the impact?
4. For CNSE, could you also report a rollout setting where DPOT is closer to its original usage (e.g., shorter rollout steps or per-step prediction with consistent teacher-forcing) to better contextualize results?

---

> ### Author Response · Authors · 2025-11-21
> **Response1 to reviewer k7Hr**
>
> ## **Response to reviewer k7Hr**
> Thank you for the valuable feedback on our paper. We hope that our responses will address the reviewer’s questions effectively. We have incorporated the additional elements discussed during the rebuttal process into the revised paper, and all newly added or fixed content is highlighted in blue text. If further clarification or additional experiments would be helpful, we would be happy to provide them.
>
> ### **[Weakness 1]**
> We agree that incorporating complex PDEs or geometries with strong stiffness would further strengthen the claim toward “foundation” capabilities. In line with this direction, we have already conducted experiments on complex PDEs, such as the compressible Navier–Stokes equations (CNSE), as well as real-world scenarios with geometric obstacles, such as the Airfoil dataset, which features nontrivial geometries. In addition, through the experiments conducted during the rebuttal period, we verify the feasibility of extending our model to 3D PDEs. Detailed information on these results can be found in our **response to Reviewer AmpV’s Weakness 1**.
>
> Beyond solving a more complex PDEs, our goal is to design a model that also generalizes across tasks and remains robust to noisy priors, rather than focusing solely on diverse PDE generalization. We believe these are equally essential attributes of a scientific foundation model (SFM). Extending this framework toward complex-geometry generalization, as you suggest, is a important next step, and we consider this an exciting direction for continued work.
>
> ### **[Weakness 2]**
>
> As you told, it is true that *vanilla* PINNs are not universally cheap or stable, and their training cost and failure modes can vary significantly across PDEs, boundary conditions, and initial conditions. Our primary motivation for using a PINN prior is not to claim that PINNs are inherently efficient, but rather to use them as a realistic source of noisy surrogate data, occasionally biased approximations that frequently arise in practical scientific workflows.
>
> In many real-world scenarios, researchers already have access to low-quality solutions, obtained from coarse solvers, long before a high-fidelity numerical simulation is available. PINN priors served as a concrete example of such noisy approximations, allowing us to systematically test whether our PPD-based in-context inference remains stable when the prior is noisy.
>
> We emphasize that our method does not depend on PINNs specifically. If higher-quality or more efficient solutions are available, they can be used directly as priors. Moreover, for PDEs where *vanilla* PINNs exhibit instability, more advanced variants such as Piratenets [1] or lightweight coarse-grid numerical solvers could be used to generate priors with no change to our framework.
>
> We confirm the robustness of our model when using PINN priors. Moreover we conduct additional experiment to further validate this robustness by injecting biased Gaussian noise to numerical prior during the inference stage. Detailed results can be found in our **response to Question 3**.
>
> Finally, regarding the total computational cost of generating PINN priors for the 2,187 CDR equations, we provide the details in our **response to Question 1**.
>
> [1] Wang, Sifan, et al. "Piratenets: Physics-informed deep learning with residual adaptive networks." Journal of Machine Learning Research 25.402 (2024): 1-51.

---

> ### Author Response · Authors · 2025-11-21
> **Response2 to reviewer k7Hr**
>
> ### **[Weakness 3]**
>
> Thank you for this helpful comment. We agree that a clearer positioning relative to recent in-context operator learning SFMs such as **ICON-LM** and **OmniArch** help clarify the distinct contributions of our work.
> While these models share high-level inspiration from in-context learning, they differ in several fundamental aspects.
>
> **ICON-LM**. As noted in the paper, ICON-LM requires task-specific demos, paired (input, output) examples of the target operator, to be included directly in the context. In contrast, our model does not require such demos; it performs zero-shot inference using only input points as context without requiring any task-level examples. Furthermore, ICON-LM often incorporates explicit PDE-form information when constructing prompts for new tasks, whereas our method does not rely on access to governing equations at all.
>
> **OmniArch**. OmniArch is capable of handling 1D–3D scientific data but requires regular grids and processes them through hierarchical self-attention. This imposes a fixed data layout and constrains the representation to a predefined operator structure established during training. Our model, by contrast, operates on both grid-based and irregular geometries, and leverages both self- and cross-attention, enabling prediction at arbitrary query locations. This design allows our method to generalize not only across PDEs but also across tasks, without task-specific fine-tuning.
>
> Finally, while ICON-LM and OmniArch focus on empirical ICL performance, neither framework provides theoretical guarantees of robustness to unbiased noise in the prior data. Our method leverages the PFN architecture to directly approximate the posterior predictive distribution (PPD), and our analysis (Theorem 2.1) shows that this leads to a form of Bayesian consistency under noisy priors.
>
> ### **[Question 1]**
>
> For the $2,187$ CDR equations, we seperately generated PINN-generated priors (PINN piror) for each equation. As described in Appendix C, generating the PINN prior for one equation takes approximately $373$ seconds, so producing all $2,187$ priors requires roughly $373 \times 2187$ seconds in total. We acknowledge that this preprocessing stage is computationally nontrivial. However, it is not part of the PDE-PFN training pipeline. Instead, it serves purely as a data generation step to create a noisy prior, and these priors are generated once solely for the purpose of evaluating robustness to noisy prior.
>
> Regarding the PDE-PFN training cost: training on the CDR family requires 551.51 seconds per epoch, while DPOT requires 423.26 seconds per epoch under same experimental conditions. Although PDE-PFN incurs a higher per-epoch cost, this is because a single training enables the model to handle multiple tasks without task-specific retraining. In contrast, neural operator baselines typically require separate training runs for each task, thereby increasing total training cost across tasks.
>
> Consequently, while prior generation was computationally expensive in this particular experimental setting, this expense is external to our model and reflects our choice to evaluate robustness across a broad and systematically varied prior space, rather than a limitation of PDE-PFN itself.

---

> ### Author Response · Authors · 2025-11-21
> **Response3 to reviewer k7Hr**
>
> ### **[Question 2]**
>
> In response to your question, we conduct an empirical analysis of how both training and inference scale with respect to (i) the number of context points and (ii) and the number of query points. changes in spatial or temporal resolution naturally correspond to changes in these two quantities.
>
> When measuring scaling behavior, we fix one of the two axes (either context or query size) to 4,096 points. This value provides a representative midpoint for evaluating scaling for experiments in main paper. We report the average runtime over 10 iterations together with peak GPU memory usage.
>
> |Context length|1,024|2,048|4,096|8,192|12,288|16,384|
> |-|-|-|-|-|-|-|
> |Training time (sec)   |0.102|0.102|0.101|0.107|0.136|0.165|
> |Training memory (MB) |280.0|329.8|420.7|617.2|800.4|987.4|
> |Inference time (sec) |0.030|0.041|0.065|0.115|0.165|0.213|
> |Inference memory (MB)|237.8|312.0|459.7|755.7|1054.7|1349.7|
>
> |Query length|1,024|2,048|4,096|8,192|12,288|16,384|
> |-|-|-|-|-|-|-|
> |Training time (sec)   |0.101|0.100|0.101|0.123|0.148|0.176|
> |Training memory (MB) |872.7|1010.1|1265.4|1788.6|2303.0|2831.2|
> |Inference time (sec) |0.056|0.059|0.065|0.078|0.090|0.104|
> |Inference memory (MB)|358.3|391.7|458.7|594.7|733.7|867.7|
>
> Across both tables, we observe that runtime and memory increase in an approximately linear fashion as the number of context or query points grows. This aligns with the theoretical behavior reported in the AGF paper [2], where attention complexity scales linearly with sequence length. While some deviation from perfect linearity appears at very large sequence sizes due to the components other than AGF, the overall trend remains close to linear. We have incorporated this empirical scaling analysis into **Appendix H.9** of the revised paper.
>
> [2] Wi, Hyowon, Jeongwhan Choi, and Noseong Park. (2025). Learning Advanced Self-Attention for Linear Transformers in the Singular Value Domain.
>
> ### **[Question 3]**
>
> To evaluate the model’s sensitivity to biased priors, we conduct an additional experiment on the 2D CDR temporal extrapolation task. During inference, we injected element-wise biased Gaussian noise into the context values, where the noise takes the form $\mathcal{N}(\mu, \sigma=0.01)$ and the bias magnitude $\mu$ is varied from 0% to 4%. The results are shown below:
>
> |Bias $\mu$|No noise|0.1%|0.5%|1%|1.5%|2%|2.5%|3%|3.5%|4%|
> |-|-|-|-|-|-|-|-|-|-|-|
> |$L\_1$ abs error |0.01474|0.01483|0.01472|0.01481|0.01514|0.01572|0.01664|0.01800|0.01985|0.02212|
> |$L\_2$ rel error |0.02261|0.02271|0.02246|0.02244|0.02271|0.02330|0.02426|0.02573|0.02772|0.03009|
> |$L\_\infty$ rel error |0.08444|0.08556|0.08459|0.08426|0.08490|0.08646|0.08901|0.09265|0.09733|0.10276|
>
> We observe that bias levels up to approximately 2.0% have only mild degradation on model performance, while larger biases produce more noticeable deterioration. This indicates that PDE-PFN retains meaningful robustness against moderately biased in the prior.
>
> We have incorporated this experiment and discussion into **Appendix H.8** of the revised manuscript.

---

> ### Author Response · Authors · 2025-11-21
> **Response4 to reviewer k7Hr**
>
> ### **[Question 4]**
> Thank you for the suggestion. We originally evaluated DPOT using a single-shot prediction over the entire time window because its publicly released checkpoint was pretrained on this dataset using per-step prediction. Therefore, using the exact same rollout strategy as in its pretraining would not constitute a fair comparison with our model or the other baselines.
>
> Nonetheless, following your recommendation, we additionally evaluated two rollout variants of DPOT on the CNSE task: rollout without fine-tuning, and rollout with fine-tuning. The results are summarized below:
>
> ||Ours|DPOT w.o. fine-tuning|DPOT with fine-tuning|
> |-|-|-|-|
> |$L\_1$ abs error |0.09497|0.51223|0.07819|
> |$L\_2$ rel error |0.54833|1.01148|0.19690|
> |$L\_\infty$ rel error |0.63533|1.01552|0.20946|
>
> As shown, fine-tuned DPOT achieves stronger performance than our model when allowed to roll out, which is consistent with its original design. However, as noted above, this is not a fair comparison, since DPOT is being evaluated under the dataset used in pretraining, whereas our model is not.
>
> DPOT's pre-trained official checkpoint was already trained with CNSE and it shows not bad performance even before fine-tuning. After fine-tuning, in addition, it shows the best performance in the table.
>
> Since our method was not pre-trained with CNSE, ours before fine-tuning is not able to solve CNSE since it was trained with CDR only. After fine-tuning, however, our method shows competitive performance for CNSE, which shows the good generalization characteristic via fine-tuning.
>
> Our method shows the nice generalization after the fine-tuning, but due to the difference that DPOT saw the dataset twice, one for pre-training and the other for fine-tuning, our method is slightly worse than "DPOT pre-trained and fine-tuned with CNSE." For completeness and fairness, we therefore reported DPOT under the new evaluation regime (single-shot prediction).

---

> ### Author Response · Authors · 2025-11-27
> **Reminder for reviewer k7Hr**
>
> ## **Reminder for reviewer k7Hr**
>
> Thank you very much for your thoughtful and constructive feedback on our submission. We sincerely appreciate the time you have taken to evaluate our work. We have carefully addressed all comments and prepared detailed responses to each question. We hope that our clarifications fully resolve your concerns, and we would be happy to provide further explanations or additional experiments if needed.
>
> During the rebuttal period, we also conducted new experiments inspired by the reviewers’ questions, which further strengthen the empirical validation of our method. All newly added or revised content has been incorporated into the updated manuscript, with changes highlighted in blue for clarity.
>
> A summary of the newly added experiments is as follows:
>
> - **Main paper Section 4** : Additional baseline - Latent Neural operator(LNO)
> - **Appendix H.5** : Unseen coefficient unseen temporal domain 2D CDR equation experiment
> - **Appendix H.6** : 3D diffusion-reaction equation experiment
> - **Appendix H.7** : Ablation study
> - **Appendix H.8** : Biased-noise robustness experiment
> - **Appendix H.9** : Computational cost analysis
> - **Appendix H.10** : 2D CDR equation experiment with different initial condition
>
> Thank you again for your time and effort in reviewing our paper. Please feel free to let us know if you have any additional questions. We would be glad to assist.

---

### Author Response · Authors · 2025-11-30
**Summary of Rebuttal Discussion for New Area Chair**

## **Summary of Rebuttal Discussion for New Area Chair**
- During the rebuttal period, we conducted a series of new experiments in direct response to reviewer questions and concerns.
- We received follow-up feedback from two reviewers, and we were able to fully address their questions and resolve their concerns.
- We initially received scores of **6/6/6/4**. After reviewing the additional experimental results, Reviewer `HzTi` explicitly increased their score **from 4 to 6**, bringing the overall scores to **6/6/6/6**. In addition, Reviewer 7z9q stated that all of their concerns had been fully addressed.

A summary of the rebuttal process for each reviewer is provided below.
### **Summary of Rebuttal for Reviewer `k7Hr`**
1. **Weakness 1**: Claims toward “foundation” capabilities would be more convincing with harder PDEs (e.g., Navier–Stokes) or complex geometries.
- **A**: We highlighted existing experiments on CNSE and Airfoil and added new 3D diffusion–reaction PDE experiment (Appendix H.6).
2. **Weakness 2 and Question 1**: PINN-based priors may be expensive or unstable, and total computational budget.
- **A**: We clarified that PINNs are used only as noisy priors (not required by our framework). We also provided a computational cost breakdown.
3. **Weakness 3**: Differences versus related SFM approaches.
- **A**: We provided detailed architectural and conceptual comparisons.
5. **Question 2**: Scaling with context/query size.
- **A**: We added a scaling analysis with linear memory/runtime behavior (Appendix H.9).
5. **Question 3**: Sensitivity to biased priors.
- **A**: We added controlled biased-noise experiments showing robustness up to 2–3% bias (Appendix H.8).
6. **Question 4**: DPOT rollout fairness for CNSE.
- **A**: We added two new DPOT rollout evaluations (fine-tuned & non-fine-tuned) and explained fairness issues due to DPOT being pretrained on CNSE.
### **Summary of Rebuttal for Reviewer `AmpV`**
1. **Weakness 1 and Question 2**: The method’s scalability to 3D or multi-physics settings is not demonstrated.
- **A**: We added the 3D diffusion–reaction PDE experiment (Appendix H.6).
2. **Weakness 2 and Question 1**: Need to isolate AGF layers, rational activations, and Fourier embeddings.
- **A**: We added extensive ablation studies (Appendix H.7).
3. **Weakness 3**: Insufficient description of sampling for PINN priors.
- **A**: We clarified uniform sampling across all 2,187 CDR equations; no class imbalance.
4. **Question 3**: AGF loss sensitivity.
- **A**: We conducted analysis showing AGF loss scaling is essential for stability. Without scaling, AGF loss dominates >90% of total loss early in training.
### **Summary of Rebuttal for Reviewer `HzTi`**
1. **Weakness 1**: Handling irregular meshes not novel enough.
- **A**: We clarified that irregular geometry support is not claimed as novelty and explained that it is only one component of the broader contribution of PDE-PFN.
2. **Weakness 2**: Are tasks truly independent? Is this really ICL or masked pretraining?
- **A**: We clarified Bayesian ICL interpretation and emphasized that PDE-PFN performs inference conditioned on arbitrary context sets, unlike operator-learning baselines.
3. **Weakness 3**: Baseline coverage limited.
- **A**: We added LNO (transformer-based) as a strong additional baseline.
4. **Weakness 4**: Need ICL evidence beyond coefficient interpolation/extrapolation.
- **A**: We added unseen coefficient + unseen temporal domain experiment (Appendix H.5).
5. **Question 1**: Experimental setting of Section 4.2.
- **A**: We clarified that all baselines received exactly the same context coordinates and solution values as PDE-PFN, ensuring fully fair and identical conditioning across models.
6. **Questions 2 and 3**: Initial condition of 2D CDR equations.
- **A**: All equations use the same IC $1+sin(2x)sin(2y)$, ensuring differences arise solely from PDE coefficients.
7. **Question 3**: Meaning of “minimal fine-tuning” and comparison to FNO.
- **A**: Minimal fine-tuning uses half the baseline training epochs; under this regime, pretrained FNO underperforms its from-scratch version.
8. **Additional Question 1**: Are trajectories unique for different PDE parameters?
- **A**: We provided an analytic explanation. Verified with the full dataset that no two PDE instances share identical contexts.
9. **Additional Question 2**: Would the method still work with different initial conditions?
- **A**: We added new IC experiment (Appendix H.10), and PDE-PFN still outperforms the baselines.
10. **Additional Question 3**: How were operator baselines trained?
- **A**: All baselines trained with 1-step prediction, matching PDE-PFN for fairness.

### **Summary of Rebuttal for Reviewer `7z9q`**
- **Question**: If PINN priors are very inaccurate, can PDE-PFN still learn effectively?
- **A**: We clarified PINNs are only one example of noisy priors; framework is not dependent on them. We also added biased-noise robustness experiment (Appendix H.8).

---

### Meta-Review · Area_Chair_hRpf · 2026-01-11

**Summary:**

The paper introduces PDE-PFN, a Transformer-based model that can be queried at arbitrary space–time coordinates to infer solutions of partial differential equations (PDEs). The proposed architecture and training methodology are inspired by prior work (Müller et al., 2022, cited in the paper), originally developed for learning posterior predictive distributions in regression and classification settings. In that framework, the model is trained on multiple datasets, each consisting of input–output pairs sampled from a prior distribution. For each dataset, the model learns to predict a held-out example given the remaining input–output pairs as context. At inference time, the model is provided with a new dataset and a query input and must predict the corresponding output. This paradigm is related to meta-learning and to in-context learning.

PDE-PFN follows a similar scheme in the context of PDE modeling. The model is trained on data simulated from a parametric family of equations formed as linear combinations of convection, diffusion, and reaction (CDR) terms with varying coefficients. During training, the model is provided with trajectories generated from different instances of these parametric equations and is tasked, for each trajectory, with predicting the system state at selected query space–time points. The authors further introduce the idea of using physics-informed neural networks (PINNs) as weak solvers to generate approximate, noisy training data, instead of relying exclusively on trajectories obtained from classical numerical solvers. At inference time, the model receives a new trajectory together with a set of query coordinates and is expected to infer the corresponding state values. The approach is evaluated on the CDR equation under both in-distribution and out-of-distribution parameter regimes, and additional experiments investigate fine-tuning the model on new PDE datasets.

The reviewers highlight several strengths of the paper, including the methodological novelty, the flexibility of the approach in handling irregularly spaced observations, and its ability to generalize to new parameter regimes and different PDEs. They also identify weaknesses, such as missing evaluations on more complex datasets, limited ablation studies, the lack of comparison with state-of-the-art neural operators on irregular meshes.

**Reviewer Concerns:**

The authors provide a strong and detailed rebuttal, addressing many of these points and presenting additional experiments on new datasets, including a 3D dataset, as well as results with an additional operator-learning baseline. Overall, these additions clearly strengthen the empirical section of the paper.

I agree with the reviewers regarding the originality and interest of the proposed approach, and I find that the new experimental results substantially improve the paper. However, I remain uncomfortable with several aspects of the presentation and interpretation. First, the paper contains a number of incorrect or overstated claims concerning novelty, which could likely be addressed through more positioning with respect to existing work. More importantly, the paper places strong emphasis on a Bayesian interpretation inspired by Müller et al. (2022), and frames its theoretical analysis in terms of posterior distributions and convergence properties. In my view, this theoretical framework is only loosely connected to the actual algorithmic setting. The proposed model is trained as a deterministic regressor using a mean-squared error objective, produces point estimates rather than posterior distributions, and does not model uncertainty. As a result, the theoretical analysis appears to provide intuition rather than a foundation for the proposed method, and the connection between the Bayesian motivation and the implemented algorithm remains unclear.

Finally, while the experimental results are overall strong, certain design choices raise questions about the fairness of some comparisons, particularly across methods with different training or inference protocols.

Overall, I consider this to be an original contribution with significant potential impact. However, in its present form, I do not believe the paper is ready for publication. It would greatly benefit from a careful revision. In particular, clarifying the scope of the theoretical claims, revising some of the novelty statements, and improving the alignment between the conceptual framing and the methodology would substantially strengthen the paper and allow its merits to be more clearly and convincingly presented.

**Reviewer Scores:**

RHzTi, rating 4, says that they will increase their score

All the other reviewers have rating 6 and will probably not change their score.

---

### Decision · Program_Chairs · 2026-01-26

Reject